

# CVPM 1.1: a flexible heat-transfer modeling system for permafrost

Gary D. Clow

Institute of Arctic and Alpine Research, University of Colorado, Boulder, Colorado, USA

**Correspondence:** Gary D. Clow (gary.clow@colorado.edu)

**Abstract.** The Control Volume Permafrost Model (CVPM) is a modular heat-transfer modeling system designed for scientific and engineering studies in permafrost terrain, and as an educational tool. CVPM implements the nonlinear heat-transfer equations in 1-D, 2-D, and 3-D cartesian coordinates, as well as in 1-D radial and 2-D cylindrical coordinates. To accommodate a diversity of geologic settings, a variety of materials can be specified within the model domain, including: organic-rich materials, sedimentary rocks and soils, igneous and metamorphic rocks, ice bodies, borehole fluids, and other engineering materials. Porous materials are treated as a matrix of mineral and organic particles with pore spaces filled with liquid water, ice, and air. Liquid water concentrations at temperatures below 0°C due to interfacial, grain-boundary, and curvature effects are found using relationships from condensed matter physics; pressure and pore-water solute effects are included. A radiogenic heat-production term allows simulations to extend into deep permafrost and underlying bedrock. CVPM can be used over a broad range of depth, temperature, porosity, water saturation, and solute conditions on either the Earth or Mars. The model is suitable for applications at spatial scales ranging from centimeters to hundreds of kilometers and at timescales ranging from seconds to thousands of years. CVPM can act as a stand-alone model, the physics package of a geophysical inverse scheme, or serve as a component within a larger earth modeling system that may include vegetation, surface water, snowpack, atmospheric or other modules of varying complexity.

## 1 Introduction

Given the recent surge of interest in the cryosphere and its role in the Earth's climate system, a large number of permafrost models have been developed over the past few decades. An important characteristic of permafrost, especially in its fine-grained form, is that significant amounts of liquid water can coexist with ice within the pore spaces at temperatures well below 0°C due to a combination of interfacial, grain-boundary, curvature, solute, and pressure effects (Davis, 2001; Dash et al., 2006). Even at standard atmospheric pressure, liquid water has been observed at temperatures as low as −31°C in silty soils and −40°C in glass powders (Watanabe and Mizoguchi, 2002). The existence of liquid water at such low temperatures is of interest biologically (Rothschild and Mancinelli, 2001), particularly in the case of Mars where permafrost could serve as a microbial refuge from high radiation levels if life ever existed there. From a geoscience perspective, it too is the presence of liquid water that makes permafrost so dynamic and interesting. Since the liquid water content of permafrost is highly temperature dependent and the thermal properties of solid and liquid water are so different (Anderson et al., 1973; Holten et al., 2012; Huber et al., 2012; Yen, 1981), the thermal response of permafrost to any temperature change is complicated by nonlinearities



and feedbacks. As with the thermal properties, the mechanical properties of permafrost can be highly sensitive to temperature and the unfrozen water content, particularly within a few degrees of the freezing point $T_F$. As temperatures approach $T_F$, the material strength generally declines increasing the likelihood of downslope creep, slope failures, accelerated lakeshore and coastal erosion, and ultimately thaw settlement (thermokarst) if temperatures become warm enough. If enough liquid water

is available and the permafrost is sufficiently permeable, migration of liquid water towards colder temperatures can lead to significant frost heave, damaging buildings, roadways, and other facilities. With a warming climate, the dynamic response of permafrost is expected to be amplified, leading to accelerated landscape changes, disruption of vulnerable habitats and ecosystems, and damage to human infrastructure (USARC, 2003; ACIA, 2005).

A wide range of models have been developed to better understand the occurrence of permafrost and its dynamics in a warm-

ing world. These models range from simple analytical models to sophisticated numerical codes with integrated vegetation, snow, and atmospheric layers overlying permafrost (e.g., Zhang et al., 2003; Riseborough et al., 2008). The vast majority of these are 1-D vertical models. Although useful for simulating conditions beneath a uniform surface, 1-D models ignore important lateral heat-transfer effects occurring near large land-surface contrasts such as at the boundary between tundra, rivers, lakes, oceans, glaciers, and human infrastructure. In addition, these models almost universally use empirical equations to predict the

unfrozen water content at temperatures below $0°C$. A significant limitation with this approach is that the coefficients appearing in the unfrozen water equations must be 'calibrated' using field data for every material type, pressure, water saturation, and solute condition. Even with calibration, the empirical equations remain valid only over a limited range of temperatures. With an emphasis on simulating shallow permafrost and active-layer conditions, most permafrost models currently neglect the freezing-point depression due to pressure and the radiogenic heat-source term, both of which are needed to simulate conditions

in deep permafrost.

In this paper, we present the new Control Volume Permafrost Model (CVPM v1.1) which is designed to relax several of the limitations imposed by previous models. CVPM implements the nonlinear heat-transfer equations in 1-D, 2-D, and 3-D cartesian coordinates, as well as in 1-D radial and 2-D cylindrical coordinates. A variety of materials can be specified within the modeling domain, including: organic-rich materials, sedimentary rocks and soils, igneous and metamorphic rocks, ice

bodies, borehole fluids, and other engineering materials. Numerical implementation is based on the control-volume method (Anderson et al., 1984; Minkowycz et al., 1988; Patankar, 1980), allowing enthalpy fluxes to be exactly balanced at control-volume interfaces (e.g., at the interface between an ice lens and a siltstone). The unfrozen water content at temperatures below $0°C$ is found using relationships from condensed matter physics that utilize physical quantities (e.g., particle radii), rather than non-physical empirical coefficients requiring calibration. Pore pressure and solute effects are included in the unfrozen

water equations. A radiogenic heat-production term is also included to allow simulations to extend into deep permafrost and underlying bedrock. CVPM is designed for use over a broad range of depth, temperature, rock and soil types, porosity, water saturation, and solute conditions. These conditions include the coldest temperatures experienced on Earth through the ice ages, as well as conditions on Mars where the upper crust of the planet consists entirely of permafrost (Squyres et al., 1992). The model is suitable for applications at spatial scales ranging from centimeters to hundreds of kilometers and at timescales ranging

from seconds to thousands of years. To achieve the greatest flexibility, CVPM does not include heat-transfer processes within



a vegetation canopy, snowpack, or atmospheric boundary layer. Rather, CVPM focuses on permafrost and the underlying earth materials. In this way, CVPM can act as a stand-alone model, the physics package of a geophysical inverse scheme, or serve as a component within a larger earth modeling system that may include vegetation, surface water, snowpack, atmospheric or other modules of varying complexity.

## 2  Governing equations

The basis for the CVPM model is the conservation of mass and enthalpy over time within any finite volume $V$. In integral form, the conservation equations take the form,

$$\int_V \frac{\partial \rho}{\partial t}\, dV = -\int_A \rho \boldsymbol{v} \cdot d\boldsymbol{A} \qquad \text{(mass)} \tag{1}$$

$$\int_V \frac{\partial (\rho H)}{\partial t}\, dV = -\int_A \boldsymbol{J} \cdot d\boldsymbol{A} + \int_V S\, dV, \quad \text{(enthalpy)} \tag{2}$$

where $\rho$ is the bulk density, $H$ is the specific enthalpy, $\rho \boldsymbol{v}$ is the mass flux, $\boldsymbol{J}$ is the enthalpy flux, $S$ is the enthalpy production rate, $t$ is time, and $A$ is the area bounding volume $V$. For the current version of CVPM, the velocity $\boldsymbol{v}$ is assumed to be sufficiently small that the advective heat flux is negligible compared to the diffusive heat flux. In this case, the enthalpy flux is simply $\boldsymbol{J} = -k\nabla T$, where $k$ is the bulk thermal conductivity and $T$ is temperature. The medium within the model domain is assumed to consist of organic-rich materials, rocks and soils, ice bodies, and engineering materials. Porous materials are

treated as a matrix ($m$) of mineral and organic particles with pore spaces filled with liquid water ($\ell$), ice ($i$), and air ($a$). The porosity at any model location is then $\phi = \phi_\ell + \phi_i + \phi_a$ where $\phi_\ell$, $\phi_i$, and $\phi_a$ are the volume fractions of the pore's constituents.

### 2.1  Heat capacity

In permafrost, the enthalpy at temperature $T$ consists of two components, one associated with the vibrational modes of the molecular lattice and the other due to the latent heat associated with the phase change of water,

$$\rho H(T) = \rho \int_0^T c_p(T')\, dT' + \rho_\ell\, \Delta H_{\text{fus}}\, \phi_\ell(T). \tag{3}$$

Here, $c_p$ is the specific heat of the bulk material, $\rho_\ell$ is the density of liquid water, and $\Delta H_{\text{fus}}$ is the specific enthalpy of fusion for water. Differentiating Eq. (3), the volumetric heat capacity at constant pressure, defined by $C \equiv \rho(\partial H/\partial T)_P$, is given by the sum of the lattice vibration and latent-heat terms,

$$C = \rho c_p + \rho_\ell\, \Delta H_{\text{fus}} \frac{\partial \phi_\ell}{\partial T}. \tag{4}$$

Since the density of air is much less than that of the other permafrost constituents, the lattice-vibration term is well approximated by the volume-weighted sum of the specific heats of the matrix materials, liquid water, and ice,

$$\rho c_p = (1 - \phi)\rho_m c_{pm} + \phi_\ell \rho_\ell c_{p\ell} + \phi_i \rho_i c_{pi}, \tag{5}$$




assuming $\phi_a < 1$.

Below 500 K, the specific heat of most matrix minerals is strongly temperature dependent, primarily due to the energies associated with the acoustic and optical modes of vibration (Kieffer, 1979, 1980). Due to the wide variety of crystalline mineral structures, simple analytic expressions for the temperature dependence of $c_{pm}$ encompassing all minerals are unavailable.

However, the relatively smooth temperature dependence predicted by detailed models indicates the specific heat for a given mineral can be adequately represented by a 3-term Taylor expansion $\omega(T)$ over the temperature range of interest for permafrost (Fig. 1). For most minerals, the specific heat falls within the range 630–870 J kg$^{-1}$ K$^{-1}$ at 300 K and tends towards zero as $T \to 0$ K (Kittel, 1967; Kieffer, 1979; Robertson, 1988). Taylor expansions describing the temperature dependence of $c_{pm}$ for most common mineral groups are built into CVPM. The matrix specific heat is then given by $c_{pm} = c_{pm}^{\circ} \omega(T)$ where $c_{pm}^{\circ}$ is

the specific heat of the dominant minerals at a standard temperature 293.15 K.

Unlike most materials, experimental data for liquid water show an anomalous increase in specific heat ($c_{p\ell}$) with decreasing temperature. Holten et al. (2012) explained this and other peculiar behaviors of supercooled water with a thermodynamic model that assumes the existence of a liquid–liquid critical point at low temperatures. Based on their interpretation of available thermodynamic data, the liquid–liquid critical temperature $T_c$ is near 227 K. A least-squares fit to a composite of data reported

by Angell et al. (1982) below 273 K and the International Association for the Properties of Water and Steam (IAPWS) 2008 values above 273 K provides the following relationships,

$$c_{p\ell}(T) = \begin{bmatrix} a_1 + a_2 \left( \dfrac{T}{T_c} - 1 \right)^{-1}, & 235\,\text{K} < T \le 265\,\text{K} \\ \sum\limits_{i=1}^{5} b_i \left( \dfrac{T}{310\,\text{K}} - 1 \right). & 265\,\text{K} < T \le 360\,\text{K} \end{bmatrix} \tag{6}$$

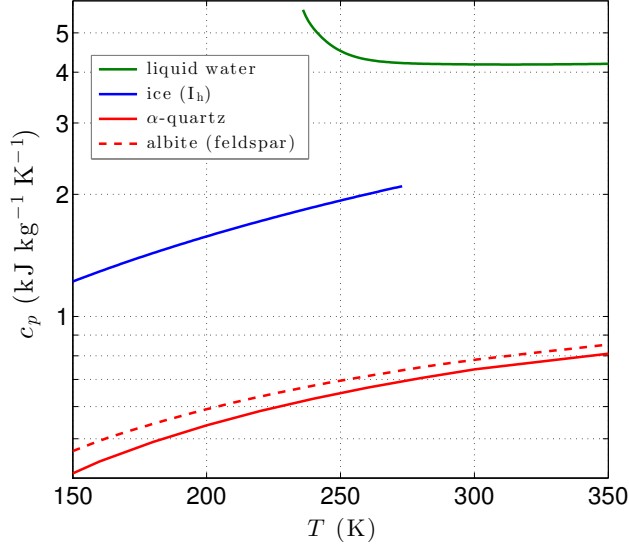

**Figure 1.** Variation of the specific heat with temperature for liquid water, ice, and two common minerals (quartz, albite).





Values for the coefficients $a_1$, $a_2$, and $b_i$ are listed in Table 1.

For water ice (I$_\text{h}$), lattice vibrations lead to a simple linear relationship between the specific heat $c_{pi}$ and temperature for $T > 150\,\text{K}$ (Yen, 1981). Based on Yen's empirical relationship, $c_{pi}$ is well-represented by,

$$c_{pi}(T) = a_1 + a_2 \left( \frac{T}{273.15\,\text{K}} - 1 \right), \quad 150\,\text{K} < T < 273.15\,\text{K} \tag{7}$$

with $a_1 = 2096.1$ and $a_2 = 1943.8$ (J kg$^{-1}$ K$^{-1}$).

## 2.2 Unfrozen water

Studies dating back to the mid-1800s show that a melt layer can stably exist at the interface between ice and a foreign substrate (e.g., a mineral grain), even at temperatures well below the bulk freezing temperature of water $T_f$ (Dash et al., 1995; Dash, 2002). A melt layer can similarly exist at the grain-boundaries within polycrystalline ice. For both *interfacial* and

*grain-boundary melting*, the liquid phase exists because it reduces the system's total free energy. Electrostatic interactions in molecular substances such as ice tend to be dominated by nonretarded van der Waals forces. In this case, the thickness of the liquid layer adjacent to a planar substrate is $d = \lambda \Delta T^{-1/3}$ where $\Delta T = (T_f - T)$ is the temperature below the bulk freezing point (Wettlaufer and Worster, 1995; Dash et al., 2006). A ramification of this behavior is that the melting and freezing of ice in contact with mineral grains occurs over a range of temperatures, rather than at a distinct temperature. Depending on the

value for the interfacial melting parameter $\lambda$, substantial amounts of liquid water can exist at temperatures well below $T_f$. For a planar substrate, the interfacial melting parameter is given approximately by,

$$\lambda = \left( \frac{2\sigma^2 \Delta\gamma T_f}{\rho_i \Delta H_\text{fus}} \right)^{1/3}, \tag{8}$$

where $\Delta\gamma$ is the difference in the interfacial free energy with and without the melt layer and $\sigma$ is a constant on the order of a molecular diameter (Wettlaufer and Worster, 1995). Imperfections due to internal disorder (polycrystallinity, point defects,

dislocations) within the ice and irregularities (pits, scratches, steps) in the substrate's surface can greatly increase the magnitude of $\Delta\gamma$ and thereby the effective interfacial melting parameter $\lambda$. Due to the irregular nature of mineral surfaces and the likely

**Table 1.** Coefficients $a_i$ and $b_i$ in Eqs (6–7) for the specific heat of liquid water and of ice ($c_{p\ell}$ and $c_{pi}$ in J kg$^{-1}$ K$^{-1}$).

| $i$ | $c_{p\ell}$ $a_i$ | $b_i$ | $c_{pi}$ $a_i$ |
|---|---|---|---|
| 1 | 3791.4 | 4178.9 | 2096.1 |
| 2 | 75.457 | 2.2374 | 1943.8 |
| 3 |  | 1509.5 |  |
| 4 |  | -7129.5 |  |
| 5 |  | 19923 |  |





disorder within interstitial ice, reliable expressions for parameter $\lambda$ are currently lacking for most earth materials. Thus, $\lambda$ is best determined experimentally for frozen ground.

Surface curvature also affects the interfacial free energy and hence the thickness of liquid water films surrounding mineral grains. By considering the detailed effects of curvature along with interfacial and grain-boundary melting, Cahn et al. (1992)

found that the volume fraction of liquid water in a porous medium consisting of spheres with radius $r$ can be described by the sum of two temperature-dependent terms,

$$\phi_\ell = a_1 \left( \frac{\lambda}{r\Delta T^{1/3}} \right) + a_2 \left( \frac{\xi}{r\Delta T} \right)^2. \tag{9}$$

The first term is due to the combined effects of interfacial melting at the surface of the spherical particles and grain-boundary melting within polycrystalline pore ice. The second term is associated with high-curvature areas on the ice-liquid interface (e.g.,

near mineral grain contact points and where ice-grain boundaries approach mineral surfaces). Coefficients $a_1$ and $a_2$ depend on how the particles are packed. For simple cubic packing, $a_1 = 1.893$ and $a_2 = 3.367$ while for cubic close packing, $a_1 = 2.450$ and $a_2 = 8.572$ (Cahn et al., 1992). Earth materials are likely to have intermediate values for the packing coefficients. In addition to its dependence on particle size and temperature, the second term depends on the interfacial free energy at the ice-water interface $\gamma_{s\ell}$. The curvature coefficient at this interface, $\xi = \gamma_{s\ell} T_f / (\rho_i \Delta H_{\text{fus}})$, numerically evaluates to $0.0259\,\mu\text{m}\,\text{K}$

(Cahn et al., 1992). Given the temperature dependencies, the interfacial/grain-boundary term ($\Delta T^{-1/3}$) dominates at large $\Delta T$ values while the second term ($\Delta T^{-2}$) dominates as temperatures approach the bulk freezing point $T_f$. As the particle size $r$ decreases, the transition between the two behaviors shifts to larger $\Delta T$ values (colder temperatures). While the first term is inversely proportional to $r$, the second term has an even stronger dependence on particle size ($\phi_\ell \propto r^{-2}$). Both terms result in greater amounts of unfrozen water in fine-grained materials (Fig. 2). Experimental data with graphitized carbon black and

polystyrene powders confirm the form of Eq. (9) for monosized particles (Cahn et al., 1992).

Although the particle and associated pore-size distributions in sandstones, limestones, and other rocks are often unimodal, those in mudrocks and soils typically are not (e.g., Kuila and Prasad, 2013). To accommodate a multimodal pore-size distribution, CVPM finds the total liquid water content by summing the contributions from the dominant modes. This is implemented in CVPM using a variant of Eq. (9),

$$\phi_\ell = \sum_j \Psi_j \left[ a_1 \left( \frac{\lambda}{r_j \Delta T^{1/3}} \right) + a_2 \left( \frac{\xi}{r_j \Delta T} \right)^2 \right], \tag{10}$$

where $\Psi_j$ is the relative volume fraction of pores associated with the mode whose mean particle size is $r_j$. Tests with sample pore-size distributions show the larger pore-sizes contribute by far the most to the unfrozen water content. Thus, CVPM currently limits the number of modes contributing to $\phi_\ell$ to two ($j \leq 2$). In this case,

$$\Psi_1 = \frac{\left(\frac{r_1}{r_2}\right)^3}{\left(\frac{r_1}{r_2}\right)^3 + \left(\frac{n_2}{n_1}\right)}, \qquad \Psi_2 = \frac{\left(\frac{n_2}{n_1}\right)}{\left(\frac{r_1}{r_2}\right)^3 + \left(\frac{n_2}{n_1}\right)} \tag{11}$$

where $(n_2/n_1)$ is the ratio of the number density of pores with radius $r_2$ to those with radius $r_1$. As an example, suppose there are 100 times as many small pores (mean modal radius $0.1\,\mu\text{m}$) as large pores (mean modal radius $2\,\mu\text{m}$). Despite the



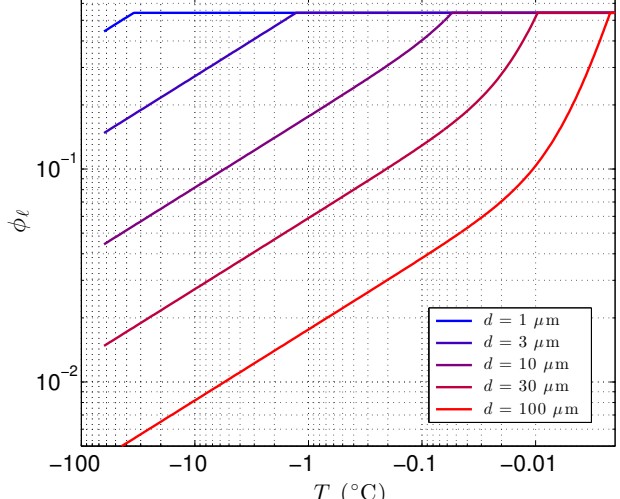

**Figure 2.** Sensitivity of the volume fraction of liquid water $\phi_\ell$ to particle diameter $d$ with $\lambda = 0.36\,\mu\mathrm{m}\,\mathrm{K}^{1/3}$ and $T_f = 273.15\,\mathrm{K}$. The porosity, $\phi = 0.54$ in this example, sets the upper limit on $\phi_\ell$.

greater number of small pores, the relative volume fraction of large pores is $\Psi_1 = 0.988$ while that of the smaller pores is only $\Psi_2 = 0.012$.

The temperature difference $\Delta T$ used to evaluate the interfacial, grain-boundary, and curvature effects is measured relative to the bulk freezing temperature $T_f$. For permafrost, pore pressures and dissolved solutes can significantly reduce $T_f$ below the point at which pure water freezes $T_f^*$ (273.16 K at the triple point pressure $P_{\mathrm{tp}} = 611.66\,\mathrm{Pa}$). If $\theta_P$ and $\theta_s$ are the freezing point depressions due to pressure and solutes, respectively, the bulk freezing temperature is,

$$T_f = T_f^* - \theta_s - \theta_P. \tag{12}$$

When solutes remain dilute, the freezing point depression due to impurities can be approximated using simple relationships such as Blagden's law (Delapaz, 2015). However, due to the insolubility of most solutes in ice, impurities become increasingly concentrated in the pore water as permafrost freezes. As a result, solute-solute interactions become increasingly important leading to significant deviations from the ideal behavior exhibited by dilute solutions. To account for the non-ideal behavior of aqueous electrolyte solutions at higher solute concentrations, CVPM uses the relationship

$$-\ln(a_w) = \frac{\Delta H_{\mathrm{fus}}^\circ}{RT^\circ}\left(\frac{\theta_s}{T^\circ}\right) + \left(\frac{\Delta H_{\mathrm{fus}}^\circ}{RT^\circ} - \frac{\Delta c_p^\circ}{2R}\right)\left(\frac{\theta_s}{T^\circ}\right)^2, \tag{13}$$

between the water activity $a_w$ and the solute freezing point depression $\theta_s$ (Robinson and Stokes, 1959). Here, $\Delta H_{\mathrm{fus}}^\circ$ is the molar enthalpy of fusion at a standard temperature $T^\circ$, $\Delta c_p^\circ$ is the difference between the molar heat capacities of liquid water and ice, and $R$ is the gas constant. Using established values for $\Delta H_{\mathrm{fus}}^\circ$, $\Delta c_p^\circ$, $R$, and $T^\circ$, Eq. (13) simplifies to,

$$-\ln a_w = (9.687 \times 10^{-3})\,\theta_s + (4.76 \times 10^{-6})\,\theta_s^2 \tag{14}$$





where $\theta_s$ is in kelvins. The extent to which aqueous electrolyte solutions deviate from ideal behavior varies greatly, depending on the composition of the solute. As a result, the water activity $a_w$ depends on the particular solute and its mole fraction $x_s$. Several expressions have been proposed for the water activity of non-ideal electrolyte solutions. CVPM uses the following proposed by Miyawaki et al. (1997),

$$a_w = (1 - x_s) \exp\left(\alpha x_s^2 + \beta x_s^3\right), \tag{15}$$

where the coefficients $\alpha$ and $\beta$ are solute dependent. For example, $\alpha = 1.825, 4.754, 11.859$ for NaCl, KCl, and MgCl$_2$, respectively, while $\beta = -20.78, -49.37, -404.5$ (Miyawaki et al., 1997). Since essentially all the solutes are concentrated in the aqueous solution upon freezing, the solute mole fraction at any stage during freezing or thawing is given by $x_s = x_s^\star[(\phi_\ell + \phi_i)/\phi_\ell]$ where $x_s^\star$ is the solute concentration in the limit of no pore ice ($\phi_i \to 0$). Once $x_s$ and $a_w$ have been established, the freezing point depression $\theta_s$ can be found by solving Eq. (14).

Figure 3 shows the sensitivity of the unfrozen water content $\phi_\ell$ to solutes predicted by CVPM for a medium-grained silt along with the temperature sensitivity $\partial\phi_\ell/\partial T$ needed to find the latent-heat component of the volumetric heat capacity $C$. The results show that even small amounts of solute can significantly affect $\phi_\ell$ and $\partial\phi_\ell/\partial T$. As noted by Dash et al. (2006), the great sensitivity of $\phi_\ell$ to impurities is a likely cause for the considerable disagreement between the results of various unfrozen water experiments. An additional sensitivity can occur at cold temperatures if solute concentrations are sufficiently high. This occurs when the solution reaches its saturation limit, beyond which the solute begins to precipitate upon further cooling. This leads to the spike in $\partial\phi_\ell/\partial T$ values near $-21°$C seen Fig. 3b when aqueous NaCl concentrations $x_s^\star$ exceed 0.005. The temperature at which the solute-saturation spike occurs varies, depending on the particular solute. Outside of the saturation limit, the largest $\partial\phi_\ell/\partial T$ values occur as the last bits of pore ice melt upon warming ($\phi_i \to 0$). The volumetric heat capacity mirrors the temperature sensitivity $\partial\phi_\ell/\partial T$ but with minimum values established by the lattice-vibration term in Eq. (4).

As previously mentioned, the interfacial melting parameter is best determined experimentally for natural earth materials. Inversion of unfrozen water data from Yuanlin and Carbee (1987) using CVPM yields a value of $\lambda = 0.36\,\mu\text{m K}^{1/3}$ for Fairbanks silt. Other parameters determined by the inversion are $\Psi_1 \simeq 1$ and $x_s^\star = 0.0008$. Thus, the pore-size distribution for this material is approximately unimodal. In addition, trace amounts of impurities appear to have been present during the unfrozen water experiments despite efforts to eliminate them. The $\lambda$ value determined for Fairbanks silt is about 100 times that determined for liquid films adjacent to smooth metal wires (Cahn et al., 1992), testifying to the importance of mineral surface irregularities and imperfections on the interfacial free energy in natural earth materials. While more work needs to be done to quantify $\lambda$ for the range of materials expected in permafrost, preliminary inversions for sedimentary materials (e.g., Suffield silty clay and kaolinite) yield values within $\pm10\%$ of that found for Fairbanks silt. At this point, the interfacial melting parameter $\lambda$ does not appear to vary substantially amongst natural earth materials.

Given that permafrost occurs at depths in excess of 1 km in some high-latitude areas and at 3–4 km beneath the polar ice sheets (Davis, 2001; MacGregor et al., 2016), the effect of pressure can be substantial on the bulk freezing temperature $T_f$ and thereby the unfrozen water content $\phi_\ell$. If the interstitial pores are freely connected to the planet's surface, the pore pressure is equal to the *hydrostatic* pressure $P = \rho_\ell g z$ where $g$ is the acceleration of gravity and $z$ is the depth below the surface. However,





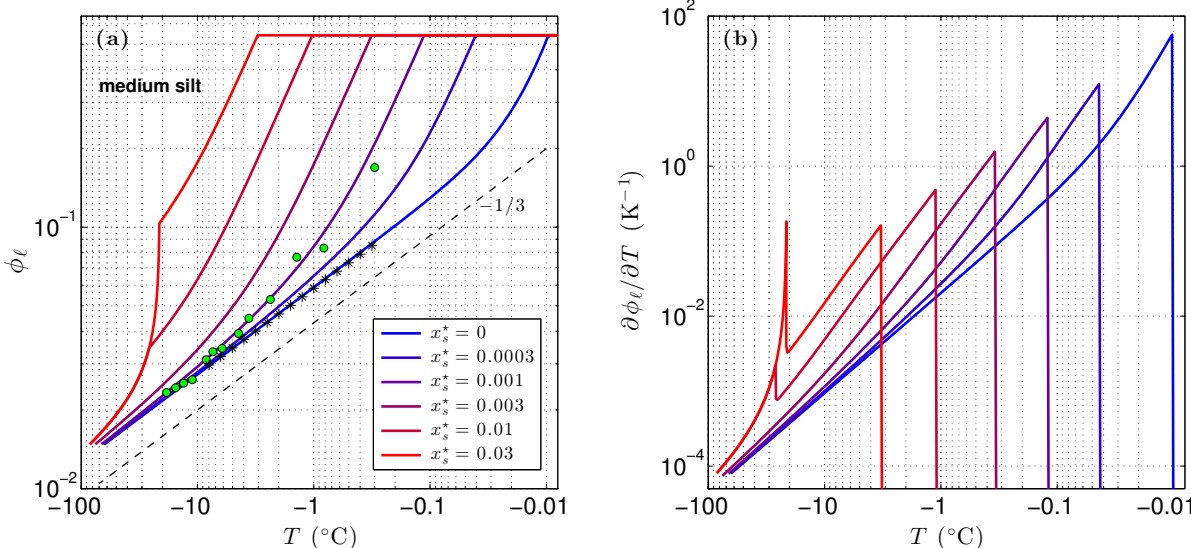

**Figure 3.** Volume fraction of liquid water $\phi_\ell$ predicted for a medium-grained silt ($r = 15\,\mu m$) with NaCl solute concentrations $x_s^\star$ ranging 0 to 0.03 (a). The depth $z \approx 0\,m$, $\phi = 0.54$, and $\lambda = 0.36\,\mu m\,K^{1/3}$. Green dots show measured $\phi_\ell$ values for Fairbanks silt ($\phi = 0.54$) reported by Yuanlin and Carbee (1987). Stars correspond to the empirical equation fitting unfrozen-water measurements in Fairbanks silt given by Anderson et al. (1973). Panel (b) shows the sensitivity of the liquid water content to temperature, $\partial\phi_\ell/\partial T$.

if the pore water is trapped, the pore pressure can be nearly equal to or exceed the *lithostatic* pressure $P = \rho g z$ (Turcotte and Schubert, 1982). In either case, the freezing point depression due to pore pressure is,

$$\theta_P = a(P - P_{\text{tp}}). \tag{16}$$

As water in permafrost is likely to be saturated with air, the appropriate value for coefficient $a$ is $9.8 \times 10^{-8}\,K\,Pa^{-1}$ (Cuffey
5 and Paterson, 2010). Since both pressure situations are known to occur in sedimentary basins, both are implemented in CVPM. The lithostatic effect is generally 2–3 times that of the hydrostatic effect. Not only does the pressure effect increase the unfrozen water content with depth, it also increases the temperature sensitivity $\partial\phi_\ell/\partial T$ and therefore the volumetric heat capacity $C$ (Fig. 4).

## 2.3 Thermal conductivity

10 Several mixing models are available for estimating the bulk thermal conductivity of multi-component systems. Of these, CVPM uses the Brailsford and Major (1964) 2-phase random mixture (BM2) and 3-phase (BM3) models recommended as being the best for use with in-situ earth materials (Roy et al., 1981). Assuming a random mixture of pores and matrix material, the bulk





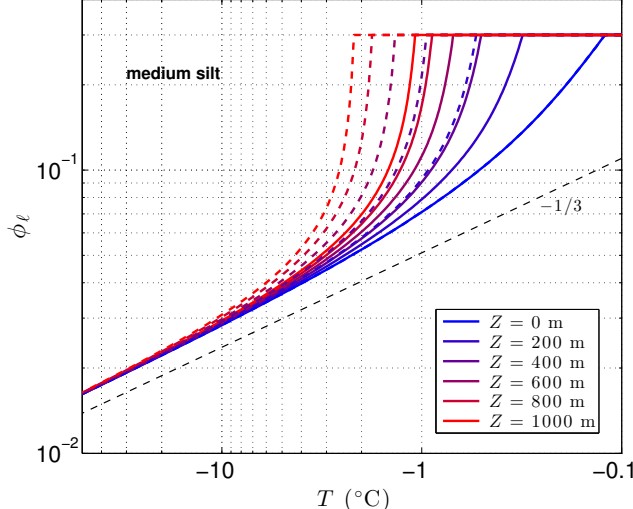

**Figure 4.** Sensitivity of the volume fraction of liquid water $\phi_\ell$ to depth below surface $z$. Solid lines are for hydrostatic pore pressures while dashed lines are for lithostatic pressures. In this example, the solute (NaCl) concentration is $x_s^\star = 0.001$, $r = 15\,\mu\mathrm{m}$, $\phi = 0.3$, and $\lambda = 0.36\,\mu\mathrm{m}\,\mathrm{K}^{1/3}$.

thermal conductivity of permafrost can be described by the BM2 model,

$$
k = k_m \left[ \frac{(2\chi - 1) - 3\phi(\chi - 1)}{4\chi} + \frac{\left\{ [(2\chi - 1) - 3\phi(\chi - 1)]^2 + 8\chi \right\}^{\frac{1}{2}}}{4\chi} \right],
$$
(17)

where $k_m$ is the conductivity of the matrix material, $k_p$ is the conductivity of the pores, and $\chi$ is their ratio $(k_m/k_p)$.

5      For matrix minerals, the thermal conductivity depends primarily on the temperature and mineral composition. Using thermal conductivity data obtained by Birch and Clark (1940a, b) over the temperature range 273–473 K, Sass et al. (1992) found the temperature dependence could be separated from the compositional dependence using a function of the form,

$$
k_m(T) = k_m^\circ \left[ a_1 + (T - 273.15\ \mathrm{K}) \left( a_2 - \frac{a_3}{k_m^\circ} \right) \right]^{-1},
$$
$$
150\,\mathrm{K} < T < 570\,\mathrm{K}
$$
(18)

10      where $k_m^\circ$ is the value of the matrix conductivity at a standard temperature 273.15 K. As the coefficients $a_i$ are fairly insensitive to rock type, the effects of mineralogy and texture are almost entirely encapsulated in $k_m^\circ$. A more recent analysis indicates the coefficients $a_i$ (Table 2) are slightly different for the mineral assemblages that dominate sedimentary rocks from those that occur in magmatic and metamorphic rocks (Vosteen and Schellschmidt, 2003). The upper temperature limit for Eq. (18) is set below the temperatures at which metamorphosis occurs in sedimentary rocks and well below the point where radiative heat





**Table 2.** Coefficients $a_i$ and $b_i$ in Eqs (18, 21–22, 24, 26–27) for the thermal conductivity of matrix minerals, liquid water, ice, air, and $CO_2$ gas ($k_x$ in W m$^{-1}$ K$^{-1}$). For matrix minerals, $a_i^{\mathrm{mm}}$ and $a_i^{\mathrm{sed}}$ refer to the coefficients appropriate for magmatic/metamorphic and sedimentary mineral assemblages, respectively.

| | $k_m$ | | $k_\ell$ | | $k_i$ | $k_a$ (air) | $k_a$ ($CO_2$) | |
|---|---|---|---|---|---|---|---|---|
| $i$ | $a_i^{\mathrm{mm}}$ | $a_i^{\mathrm{sed}}$ | $a_i$ | $b_i$ | $a_i$ | $a_i$ | $a_i$ | $b_i$ |
| 1 | 0.99 | 0.99 | 1.6630 | -1.15 | 9.828 | 0.14805 | 0.4226159 | 0.02387869 |
| 2 | 0.0030 | 0.0034 | -1.7781 | -3.4 | 0.0057 | -0.71777 | 0.6280115 | 4.350794 |
| 3 | 0.0042 | 0.0039 | 1.1567 | -6.0 | | 1.1423 | -0.5387661 | -10.33404 |
| 4 | | | -0.432115 | -7.6 | | -0.093848 | 0.6735941 | 7.981590 |
| 5 | | | | | | -1.933 | 0 | -1.940558 |
| 6 | | | | | | 2.6468 | 0 | |
| 7 | | | | | | -1.6072 | -0.4362677 | |
| 8 | | | | | | 0.48503 | 0.2255388 | |
| 9 | | | | | | -0.058451 | | |

transfer within crystal lattices becomes important (Clauser and Huenges, 1995). As very little thermal conductivity data exists for rocks and minerals below 273 K, the validity of Eq. (18) has yet to be tested at lower temperatures. The little data that does exist suggests the transition from the intermediate-temperature behavior (Eq. 18) to the low-temperature behavior $k_m \propto T^3$ (Parrott and Stuckes, 1975) generally occurs below 100 K. For example in garnets, the transition occurs at 20–30 K (Slack and Oliver, 1971). We tentatively set the lower limit of validity for Eq. (18) at 150 K.

To find the thermal conductivity of the pores $k_p$, CVPM utilizes the 3-phase BM3 model,

$$k_{\mathrm{BM3}} = k_1 \left[ \frac{\psi_1 + 3\left( \dfrac{\psi_2}{2\chi_2 + 1} + \dfrac{\psi_3}{2\chi_3 + 1} \right)}{\psi_1 + 3\left( \dfrac{\psi_2\chi_2}{2\chi_2 + 1} + \dfrac{\psi_3\chi_3}{2\chi_3 + 1} \right)} \right] \tag{19}$$

where the three phases are liquid water, ice, and air. Here, $\chi_2 = (k_1/k_2)$, $\chi_3 = (k_1/k_3)$, and the $\psi_x$ are the relative volume fractions of the pore's constituents ($\psi_x = \phi_x/\phi$). Similar to other 3-phase models, BM3 assumes phases 2 and 3 are randomly distributed within a continuous phase 1. If the relative volume fraction of any of the 3 constituents exceeds a threshold ($\psi_x \geq \alpha$), it is assumed that component is the continuous phase and $k_p$ is calculated directly from Eq. (19). A comparison of the results from the BM3 model in the limit $\psi_2 \to 0$ or $\psi_3 \to 0$ with those of model BM2 suggests a reasonable choice for $\alpha$ is $\sim 0.75$. If none of the relative volume fractions exceed $\alpha$, Eq. (19) is used to calculate the conductivity of the pore space assuming each of the 3 components, in turn, is the continuous phase to produce values $k_{c\ell}$ (continuous liquid-water phase), $k_{ci}$ (continuous ice phase), and $k_{ca}$ (continuous air phase). The pore conductivity is then found from a simple weighted average,

$$k_p = w_\ell k_{c\ell} + w_i k_{ci} + w_a k_{ca} \tag{20}$$





where the weights $w_x$ are based on the relative volume fractions $\psi_x$ and the requirement that $k_p$ be continuous across the lines $\psi_\ell = \alpha$, $\psi_i = \alpha$, $\psi_a = \alpha$ in 3-phase space (Fig. 5).

For the thermal conductivity of liquid water $k_\ell$, CVPM uses the simplified correlating equation recommended by Huber et al. (2012) for use at 0.1 MPa,

$$k_\ell(T) = \sum_{i=1}^{4} a_i \left( \frac{T}{300\,\mathrm{K}} \right)^{b_i}, \qquad 250\,\mathrm{K} < T \le 383\,\mathrm{K} \tag{21}$$

with coefficients $a_i, b_i$ (Table 2). Although the formal lower limit for Eq. (21) is 273.15 K, Huber et al. (2012) find that it extrapolates in a physically reasonable manner down to $\sim 250\,\mathrm{K}$ (Fig. 6), producing results very close to those of the new detailed IAPWS formulation for $k_\ell$. Thermal conductivity data for supercooled water does not appear to exist below 250 K at this time, preventing the development of accurate correlating equations at lower temperatures. This is a minor limitation for the thermal model as the relative amount of liquid water is expected to be small at colder temperatures.

Experimental data for the thermal conductivity of ice $k_i$ exists at temperatures as cold as 60 K. Based on this data, Yen (1981) recommends the function,

$$k_i(T) = a_1 \exp(-a_2 T), \qquad 60\,\mathrm{K} < T \le 273.15\,\mathrm{K} \tag{22}$$

for describing the temperature dependence of $k_i$, with $a_1 = 9.828\,\mathrm{W\,m^{-1}\,K^{-1}}$ and $a_2 = 0.0057\,\mathrm{K^{-1}}$.

For the terrestrial environment, the thermal conductivity of air $k_a$ can be separated into the sum of two terms, a 'dilute gas' term $k_o$ that depends solely on temperature and a 'residual' term $\Delta k$ that depends on air density,

$$k_a(\rho_a, T) = k_o(T) + \Delta k(\rho_a). \tag{23}$$

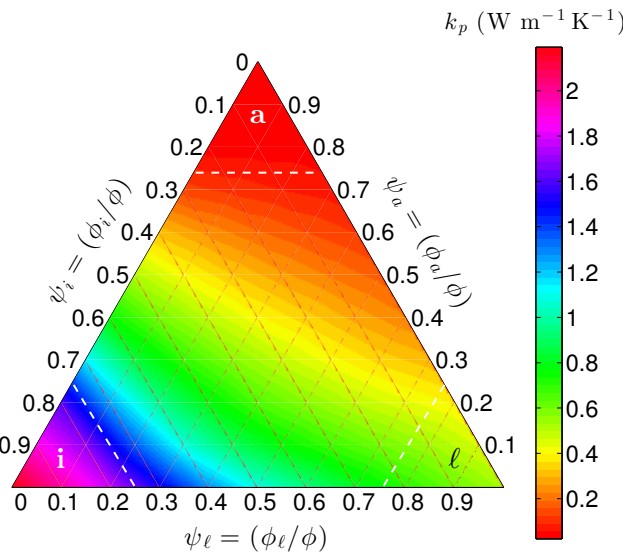

**Figure 5.** Variation of the pore thermal conductivity $k_p$ on Earth at $-10°\mathrm{C}$ with the relative volume fractions of liquid water ($\psi_\ell$), ice ($\psi_i$), and air ($\psi_a$) within the pores. Threshold $\alpha$ is 0.75 in this example.



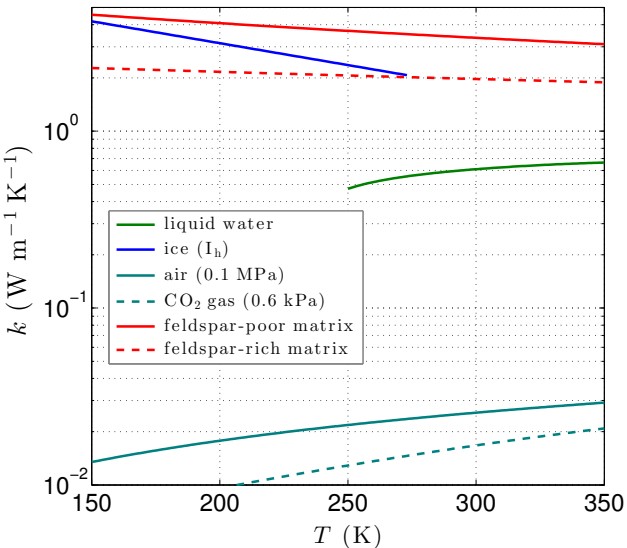

**Figure 6.** Variation of the thermal conductivity with temperature for liquid water, ice, air (terrestrial atmosphere), $CO_2$ gas (Martian atmosphere), and magmatic/metamorphic matrix minerals with $k_m^{\circ} = 3.5\,\mathrm{W\,m^{-1}\,K^{-1}}$ (feldspar-poor) and $k_m^{\circ} = 2.0\,\mathrm{W\,m^{-1}\,K^{-1}}$ (feldspar-rich).

For the dilute gas term, Stephan and Laesecke (1985) recommend the correlating equation,

$$k_o(T) = \sum_{i=1}^{9} a_i \left( \frac{T}{132.52\,\mathrm{K}} \right)^{(i-4)/3}, \quad 70\,\mathrm{K} < T < 10^3\,\mathrm{K} \tag{24}$$

with coefficients $a_i$ (Table 2). At typical terrestrial surface pressures ($\sim 0.1\,\mathrm{MPa}$), the residual term $\Delta k$ is $5.17 \times 10^{-5}\,\mathrm{W\,m^{-1}\,K^{-1}}$ (Stephan and Laesecke, 1985).

5    When considering permafrost on Mars, the thermal properties of a different atmospheric gas must be used. The Martian atmosphere is currently 95% carbon dioxide, a gas that has a thermodynamic critical point at $304.107\,\mathrm{K}$, $7.3721\,\mathrm{MPa}$. At gas densities below $25\,\mathrm{kg\,m^{-3}}$ the effects of the critical region are small enough that the thermal conductivity can again be described by Eq. (23). In the case of $CO_2$, the dilute gas contribution to $k_a$ is,

$$k_o(T) = \frac{4.75598 \times 10^{-4} \left( 1 + \dfrac{2c_{\mathrm{int}}}{5k_{\mathrm{B}}} \right) T^{1/2}}{\mathcal{C}_R(T)},$$

10    $$200\,\mathrm{K} < T < 10^3\,\mathrm{K} \tag{25}$$





where $c_{\text{int}}$ is the ideal-gas heat capacity, $k_{\text{B}}$ is the Boltzmann constant, and $\mathcal{C}_R$ is the reduced effective cross-section (Vesovic et al., 1990). The correlating equations for $\mathcal{C}_R$ and $c_{\text{int}}$ provided by Vesovic et al. (1990) are,

$$\mathcal{C}_R(T) = \sum_{i=0}^{7} a_i \left( \frac{T}{251.196\,\text{K}} \right)^{-i} \tag{26}$$

$$c_{\text{int}} = k_{\text{B}} \left[ 1 + \exp\left( \frac{-183.5\,\text{K}}{T} \right) \sum_{i=1}^{5} b_i \left( \frac{T}{100\,\text{K}} \right)^{2-i} \right], \tag{27}$$

5 (coefficients $a_i$, $b_i$ given in Table 2). At current Martian surface pressures (0.6 kPa), the residual component $\Delta k$ is 3.53 $\times 10^{-7}\,\text{W}\,\text{m}^{-1}\,\text{K}^{-1}$. Even when the Martian atmosphere was denser, the contribution of $\Delta k$ to the overall thermal conductivity of the $CO_2$ gas would have been quite small.

### 2.4 Compaction and heat-production functions

In sedimentary basins, overburden pressure causes the porosity $\phi$ to decrease with depth due to pressure solution and mechanical-
10 compaction processes (Revil et al., 2002). The former process changes the mineral shapes in response to grain-contact stresses while the latter results in the slippage and rotation of the grains. With increasing overburden pressure, the porosity ultimately reaches a residual (or critical) porosity $\phi_c$ that depends on the grain shape and grain-size distribution. Shales and mudstones are much more easily compacted than sandstones due to the plate-like shape of the mineral grains. Although fairly sophisticated compaction models now exist, CVPM uses the simple frequently-used compaction function attributed to Athy (1930),

$$\phi(z) = \phi_0 \exp\left(-z/h_c\right), \qquad \phi \geq \phi_c \tag{28}$$

to account for overburden pressures. This function has been successfully used in a large number of studies (e.g., Fjeldskaar et al., 2004; Burns et al., 2005). Here, $\phi_0$ is the porosity extrapolated to the surface while $h_c$ is the compaction length scale. Parameters $\phi_0$ and $h_c$ depend on both the lithology and the effective stress history.

CVPM assumes the enthalpy-production rate $S$ is associated with the decay of radionuclides. In this case,

$$S(z) = S_0 \exp\left(-z/h_s\right) \tag{29}$$

where $S_0$ is the surface radioactive heat-production rate and $h_s$ is the heat-production length scale (Turcotte and Schubert, 1982). Surface heat-production rates $S_0$ can vary from 0.002 to 5.5 $\mu\text{W}\,\text{m}^{-3}$, depending on lithology (Rybach, 1988), while $h_s$ is typically on the order of 10 km.

### 3 Numerical implementation

25 The CVPM modeling system implements the governing equations in 1-D, 2-D, and 3-D cartesian coordinates (X, XZ, XYZ), as well as in 1-D radial (R) and 2-D cylindrical (RZ) coordinates. Discretization follows the control-volume approach (Patankar, 1980; Anderson et al., 1984; Minkowycz et al., 1988) in which the problem domain is divided into a set of contiguous 'control





volumes' (CVs). Scalars such as temperature $T$ and thermal conductivity $k$ are computed at grid points located in the center of the CVs while the enthalpy fluxes $J$ are computed at control-volume interfaces (Fig. 7). Development of the CVPM permafrost model begins by integrating the two conservation equations (Eqs 1–2) over a time step $\Delta t$. With velocity $\boldsymbol{v} \simeq 0$, the conservation equations become,

$$
\int_V \left( \rho^{n+1} - \rho^n \right) dV = 0 \tag{30}
$$

$$
\int_V \left[ (\rho H)^{n+1} - (\rho H)^n \right] dV
$$

$$
= -\int_{t^n}^{t^{n+1}} \int_A \boldsymbol{J} \cdot d\boldsymbol{A}\, dt + \int_{t^n}^{t^{n+1}} \int_V S\, dV\, dt, \tag{31}
$$

where superscript $n$ refers to the time step following standard numerical nomenclature (e.g., $t^{n+1} = t^n + \Delta t$). Equation (30) states the bulk density integrated over any control-volume is time-invariant. The conservation of enthalpy equation (31) can be

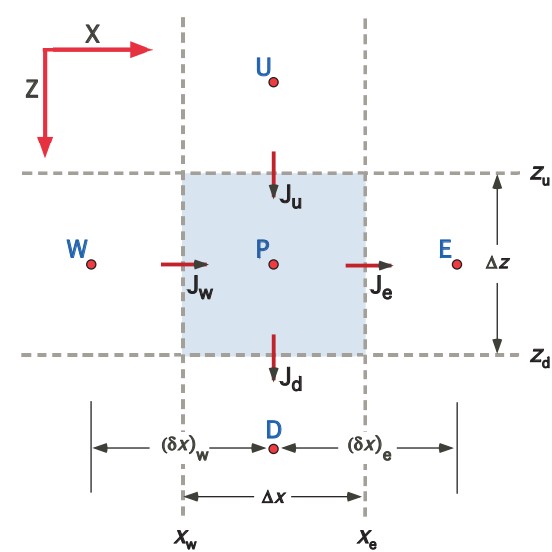

**Figure 7.** Schematic showing the nomenclature associated with a control-volume centered on grid point P for 2-D cartesian (XZ) coordinates. The control-volume is bounded by interfaces located at $x_w$, $x_e$, $z_u$, and $z_d$, through which enthalpy fluxes $J_w$, $J_e$, $J_u$, and $J_d$ pass. Grid points W, E, U, and D are located at the center of the neighboring CVs. The nomenclature for 2-D cylindrical coordinates is completely analogous with R replacing X. Three dimensional (3-D) cartesian coordinates introduces an additional axis (Y) with neighboring grid points S and N.


put in a more convenient form by noting,

$$(\rho H)^{n+1} - (\rho H)^n = \left[ \rho c_p + \rho_\ell \, \Delta H_{\text{fus}} \left. \frac{\partial \phi_\ell}{\partial T} \right|^{T^n} \right] \left( T^{n+1} - T^n \right)$$
$$= C^n \left( T^{n+1} - T^n \right). \tag{32}$$

This follows from Eq. (3) and a Taylor series expansion. To provide the flexibility of running the model in either explicit,

implicit, or fully implicit modes, the net heat flux into a control-volume is approximated by a linear combination of values at either end of a time step,

$$- \int_{t^n}^{t^{n+1}} \int_A \boldsymbol{J} \cdot d\boldsymbol{A} \, dt = -\Delta t \left[ f \left( \int_A \boldsymbol{J} \cdot d\boldsymbol{A} \right)^{n+1} \right.$$
$$\left. + (1-f) \left( \int_A \boldsymbol{J} \cdot d\boldsymbol{A} \right)^n \right]. \tag{33}$$

The explicit/implicit weighting factor $f$ can take any value between 0 and 1. Following Patankar (1980), the heat fluxes across

the $z_u$ and $z_d$ interfaces in the vertical direction (see Fig. 7) are,

$$J_u = -\tilde{k}_u \left( \frac{T_P - T_U}{z_P - z_U} \right), \quad J_d = -\tilde{k}_d \left( \frac{T_D - T_P}{z_D - z_P} \right) \tag{34}$$

where $\tilde{k}_u$ and $\tilde{k}_d$ are the 'effective' conductivities at the upper and lower interfaces, defined by,

$$\tilde{k}_u = \left[ \frac{1}{\frac{(1-\varepsilon_u)}{k_U} + \frac{\varepsilon_u}{k_P}} \right], \quad \tilde{k}_d = \left[ \frac{1}{\frac{(1-\varepsilon_d)}{k_P} + \frac{\varepsilon_d}{k_D}} \right], \tag{35}$$

with fractional distances,

$$\varepsilon_u = \left( \frac{z_P - z_u}{z_P - z_U} \right), \quad \varepsilon_d = \left( \frac{z_D - z_d}{z_D - z_P} \right). \tag{36}$$

Subscripts used here indicate grid point and interface locations. For example, $T_P$ is the temperature at grid point P while $J_u$ is the heat flux across the interface located at depth $z_u$. Fluxes across the other interfaces are defined in a completely analogous way. The use of effective conductivities guarantees that the heat fluxes exactly balance at an interface between materials with very different thermal properties (e.g., between siltstone and an ice lens). The source-term integral in Eq. (31) is left in a very

general form,

$$\mathcal{S}_P = \Delta t \int_V S \, dV. \tag{37}$$

Substituting Eqs (32–34, 37) into Eq. (31), the discrete form of the enthalpy balance for a control volume centered on grid point P can be written as,

$$a_P T_P^{n+1} = \sum a_{\text{nb}} T_{\text{nb}}^{n+1} + \sum a'_{\text{nb}} T_{\text{nb}}^n + b, \tag{38}$$



where the sums are taken over the values at the neighboring (nb) grid points (W, E, N, S, U, D). Putting all the geometric information into factors $A_x$ and $V_P$ (Table 3), the discretization coefficients for the internal control volumes are,

$$
\begin{aligned}
a_W &= f\Delta t\, A_w \tilde{k}_w, & a_E &= f\Delta t\, A_e \tilde{k}_e \\
a_S &= f\Delta t\, A_s \tilde{k}_s, & a_N &= f\Delta t\, A_n \tilde{k}_n \\
a_U &= f\Delta t\, A_u \tilde{k}_u, & a_D &= f\Delta t\, A_d \tilde{k}_d \\
a_P &= V_P C_P^n + \sum a_{\text{nb}}, & a_P' &= V_P C_P^n - \sum a_{\text{nb}}' \\
b &= \mathcal{S}_P.
\end{aligned}
\tag{39}
$$

The primed counterparts of $a_W$, $a_E$, $a_S$, $a_N$, $a_U$, $a_D$ are identical except that $f$ is replaced by $(1-f)$.

Consideration of the enthalpy balance shows that the discretization coefficients are slightly different for CVs adjacent to the boundaries of the problem domain. For a control volume adjacent to a boundary with a Dirichlet boundary condition, a factor of $(4/3)$ is introduced into the discretization coefficients (Eq. 39) associated with the boundary and the opposing interface. When the heat flux is prescribed on a boundary (Neumann BC), the coefficients associated with the boundary are zero and the specified heat flux appears in discretization coefficient $b$ (Table 4). Boundary conditions along the edges of the problem domain are allowed to vary both spatially and temporally in CVPM.

To complete the setup of the discretization coefficients, the material properties must be specified at every grid point within the model domain. These properties include: material type, mean density of matrix particles $\rho_m$, mineral-grain thermal conductivity $k_m^\circ$ at 273.15 K, mineral-grain specific heat $c_p^\circ$ at 293.15 K, heat-production rate extrapolated to the surface $S_0$, heat-production length scale $h_s$, porosity extrapolated to the surface $\phi_0$, critical porosity $\phi_c$, compaction length scale $h_c$, degree of

**Table 3.** Geometric factors $A_x$ and $V_P$ appearing in the enthalpy discretization equation (39) for cartesian and cylindrical coordinate systems. Dimensions of the control volume centered on grid point P are $\Delta x = (x_e - x_w)$, $\Delta y = (y_n - y_s)$, $\Delta z = (z_d - z_u)$ while the distance between grid points in the X direction are $(\delta x)_w = (x_P - x_W)$ and $(\delta x)_e = (x_E - x_P)$. Distances between grid points in Y and Z directions are defined similarly. For radial geometries, $\Lambda = (r_e^2 - r_w^2)/2$.

| Coordinate System | $A_w$ | $A_e$ | $A_s$ | $A_n$ | $A_u$ | $A_d$ | $V_P$ |
|---|---|---|---|---|---|---|---|
| Z | 0 | 0 | 0 | 0 | $\dfrac{1}{(\delta z)_u}$ | $\dfrac{1}{(\delta z)_d}$ | $\Delta z$ |
| XZ | $\dfrac{\Delta z}{(\delta x)_w}$ | $\dfrac{\Delta z}{(\delta x)_e}$ | 0 | 0 | $\dfrac{\Delta x}{(\delta z)_u}$ | $\dfrac{\Delta x}{(\delta z)_d}$ | $\Delta x \Delta z$ |
| XYZ | $\dfrac{\Delta y\Delta z}{(\delta x)_w}$ | $\dfrac{\Delta y\Delta z}{(\delta x)_e}$ | $\dfrac{\Delta x\Delta z}{(\delta y)_s}$ | $\dfrac{\Delta x\Delta z}{(\delta y)_n}$ | $\dfrac{\Delta x\Delta y}{(\delta z)_u}$ | $\dfrac{\Delta x\Delta y}{(\delta z)_d}$ | $\Delta x \Delta y \Delta z$ |
| R | $\dfrac{r_w}{(\delta r)_w}$ | $\dfrac{r_e}{(\delta r)_e}$ | 0 | 0 | 0 | 0 | $\Lambda$ |
| RZ | $\dfrac{r_w \Delta z}{(\delta r)_w}$ | $\dfrac{r_e \Delta z}{(\delta r)_e}$ | 0 | 0 | $\dfrac{\Lambda}{(\delta z)_u}$ | $\dfrac{\Lambda}{(\delta z)_d}$ | $\Lambda \Delta z$ |





**Table 4.** Discretization coefficient $b$ for a control volume adjacent to a prescribed heat-flux (Neumann) boundary condition.

| Coordinate System | Boundary Location | Prescribed Heat Flux | Coefficient $b$ |
|---|---|---|---|
| Z | $\min(Z)$ | $q_s(t)$ | $\mathcal{S}_P + \Delta t \left[ f q_s^{n+1} + (1-f) q_s^n \right]$ |
| | $\max(Z)$ | $q_b(t)$ | $\mathcal{S}_P - \Delta t \left[ f q_b^{n+1} + (1-f) q_b^n \right]$ |
| XZ | $\min(X)$ | $q_a(t)$ | $\mathcal{S}_P + \Delta z \, \Delta t \left[ f q_a^{n+1} + (1-f) q_a^n \right]$ |
| | $\max(X)$ | $q_o(t)$ | $\mathcal{S}_P - \Delta z \, \Delta t \left[ f q_o^{n+1} + (1-f) q_o^n \right]$ |
| XYZ | $\min(X)$ | $q_a(t)$ | $\mathcal{S}_P + \Delta y \, \Delta z \, \Delta t \left[ f q_a^{n+1} + (1-f) q_a^n \right]$ |
| | $\max(X)$ | $q_o(t)$ | $\mathcal{S}_P - \Delta y \, \Delta z \, \Delta t \left[ f q_o^{n+1} + (1-f) q_o^n \right]$ |
| | $\min(Y)$ | $q_c(t)$ | $\mathcal{S}_P + \Delta x \, \Delta z \, \Delta t \left[ f q_c^{n+1} + (1-f) q_c^n \right]$ |
| | $\max(Y)$ | $q_d(t)$ | $\mathcal{S}_P - \Delta x \, \Delta z \, \Delta t \left[ f q_d^{n+1} + (1-f) q_d^n \right]$ |
| | $\min(Z)$ | $q_s(t)$ | $\mathcal{S}_P + \Delta x \, \Delta y \, \Delta t \left[ f q_s^{n+1} + (1-f) q_s^n \right]$ |
| | $\max(Z)$ | $q_b(t)$ | $\mathcal{S}_P - \Delta x \, \Delta y \, \Delta t \left[ f q_b^{n+1} + (1-f) q_b^n \right]$ |
| R | $\min(R)$ | $q_a(t)$ | $\mathcal{S}_P + \Delta t \left[ f r_w q_a^{n+1} + (1-f) r_w q_a^n \right]$ |
| | $\max(R)$ | $q_o(t)$ | $\mathcal{S}_P - \Delta t \left[ f r_e q_o^{n+1} + (1-f) r_e q_o^n \right]$ |
| RZ | $\min(R)$ | $q_a(t)$ | $\mathcal{S}_P + \Delta z \, \Delta t \left[ f r_w q_a^{n+1} + (1-f) r_w q_a^n \right]$ |
| | $\max(R)$ | $q_o(t)$ | $\mathcal{S}_P - \Delta z \, \Delta t \left[ f r_e q_o^{n+1} + (1-f) r_e q_o^n \right]$ |
| | $\min(Z)$ | $q_s(t)$ | $\mathcal{S}_P + \Lambda \, \Delta t \left[ f q_s^{n+1} + (1-f) q_s^n \right]$ |
| | $\max(Z)$ | $q_b(t)$ | $\mathcal{S}_P - \Lambda \, \Delta t \left[ f q_b^{n+1} + (1-f) q_b^n \right]$ |

pore saturation $S_r$, dominant solute type, mole fraction of solutes extrapolated to zero ice $x_s^{\star}$, interfacial melting parameter $\lambda$, mean diameter of larger mode pores $d_1$, mean diameter of smaller mode pores $d_2$, and the ratio of the number density of small pores to large pores ($n_2/n_1$). The material 'type' specifies which governing equations to utilize when finding the heat capacity and thermal conductivity (Sect. 2). Only a subset of these properties need be specified for non-porous layers (e.g., an ice lens or a borehole casing).

Any temperature field can be used to set the initial temperature condition, including: a user-supplied field (e.g., a measured temperature field), a CVPM-determined steady-state field consistent with the boundary conditions and material properties, or a field generated by a previous CVPM modeling experiment.

With the initial condition, boundary conditions, and discretization coefficients specified, the enthalpy-balance equation (38) is solved recursively at each time step using the TriDiagonal Matrix Algorithm (TDMA) for 1-D models and the line-by-line method with TDMA for 2-D and 3-D models. Given their temperature sensitivities, the thermophysical properties ($\phi_\ell$, $C$, $k$ ...) are updated at every time step. In order for the numerical scheme to remain unconditionally stable, all of the discretization





coefficients must be non-negative. This consideration leads to the numerical stability condition,

$$\Delta t \; < \; \frac{V_P \, C_P^n}{(1-f) \sum A_{\mathrm{nb}} \tilde{k}_{\mathrm{nb}}}, \tag{40}$$

which must be satisfied in all the CVs at each time step for the scheme to remain unconditionally stable.

Model verification was conducted in two phases. In the first, the general model structure and numerical implementation were
tested by comparing model results with analytic solutions for a series of simple heat-transfer problems without phase change.
Test problems included steady-state and transient boundary conditions, homogeneous and composite media with fixed thermal
properties, materials whose thermal properties vary linearly with temperature, and materials with and without radiogenic heat-
ing. In all cases, maximum model errors $\epsilon$ are on the order of $0.1\,\mathrm{mK}$ or less under the test conditions (spatial resolution, time
step, . . . ). For most cases, $\max(\epsilon)$ ranges $1\,\mu\mathrm{K}$ to $0.01\,\mathrm{pK}$. Since analytic solutions are unavailable for simultaneously testing
all of the model physics, the second testing phase consisted of separately testing each physics module to guarantee it properly
simulates the appropriate governing equations (Sect. 2).

## 4 Example simulations with a sedimentary sequence

To illustrate the capabilities of the CVPM model, several examples are provided in this section based on the response of a 1-km
thick vertical sequence of sedimentary rocks to changing boundary conditions. The sequence consists of flat-lying mudrock,
carbonate, and sandstone units of various thicknesses (Fig. 8). Values of the parameters controlling the thermophysical proper-
ties are listed in Table 5. Heat-production rates are from Rybach (1988) while the compaction length scale is loosely derived

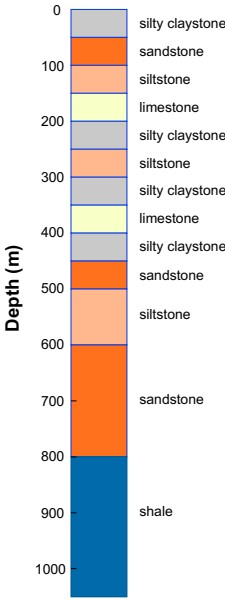

**Figure 8.** Vertical sequence of sedimentary rocks used for the permafrost example simulations.



**Table 5.** Thermophysical parameters of the sedimentary rocks used for the permafrost example simulations (Sect. 4). Units are: $k_m^\circ$ (W m$^{-1}$ K$^{-1}$), $\rho_m$ (kg m$^{-3}$), $c_{pm}^\circ$ (J kg$^{-1}$ K$^{-1}$), $S_0$ (mW m$^{-3}$), $h_s$ (km), $h_c$ (km), $\lambda$ (µm K$^{1/3}$), $d_1$ (µm), $d_2$ (µm).

| Material | $k_m^\circ$ | $\rho_m$ | $c_{pm}^\circ$ | $S_0$ | $h_s$ | $\phi_0$ | $\phi_c$ | $h_c$ | $S_r$ | $x_s^\star$ | $\lambda$ | $d_1$ | $d_2$ | $(n_2/n_1)$ |
|---|---|---|---|---|---|---|---|---|---|---|---|---|---|---|
| shale | 1.9 | 2650 | 780 | 1.8 | 10 | 0.41 | 0.05 | 1.4 | 1 | 0.003 | 0.33 | 2 | 0.1 | 1 |
| limestone | 3.7 | 2650 | 780 | 0.6 | 10 | 0.38 | 0.05 | 2.0 | 1 | 0.003 | 0.39 | 10 | — | 0 |
| silty claystone | 1.9 | 2650 | 780 | 1.8 | 10 | 0.41 | 0.05 | 1.4 | 1 | 0.003 | 0.39 | 10 | 2 | 2.55 |
| siltstone | 1.9 | 2650 | 780 | 1.8 | 10 | 0.37 | 0.05 | 2.0 | 1 | 0.003 | 0.36 | 30 | — | 0 |
| sandstone | 4.2 | 2660 | 740 | 0.8 | 10 | 0.36 | 0.10 | 2.4 | 1 | 0.003 | 0.36 | 177 | — | 0 |

from values found for a partially exhumed basin on the Arctic Slope of Alaska (Burns et al., 2005). Pores spaces are assumed to be fully saturated with water throughout the geologic section ($S_r \equiv 1 - \phi_a/\phi = 1$). Sodium chloride, present at relatively low levels when the sediments are in a thawed state ($x_s^\star = 0.003$), is the dominant pore-water solute. The conductive heat flux at the base of the model domain ($q_b = 60\,\mathrm{mW\,m^{-2}}$) is slightly above the continental average.

## 4.1 Permafrost response to ice-age cycles

The first simulation explores the response of the sedimentary sequence to surface-temperature changes over the last ice-age cycle. The upper boundary condition is based on the surface-temperature history determined for the Greenland Ice Sheet during the Holocene and Wisconsin Glacial Period by Cuffey and Clow (1997) and the Eemian interglacial by the NEEM Community Members (2013). To construct the upper BC for the permafrost simulation, the Greenlandic temperature history was rescaled and shifted to yield an 8 K warming between the Last Glacial Maximum (LGM) and the early Holocene and a mean surface temperature of $-11^\circ$C during the 1800s (Fig. 9). For the most recent period, upper boundary temperatures warm 2.5 K between the mid-1800s and 1980 and an additional 2.5 K by the present time, similar to the record for the Arctic Coastal Plain of Alaska (Clow, 2017). The temperature history was replicated back several ice-age cycles to allow for model spin-up.

Initializing the model simulation at 255 ka with the mean surface temperature over an ice-age cycle, CVPM was run forward in its 1-D vertical mode to the present time. During the last ice-age cycle, the base of permafrost $P_d$ (defined by the $0^\circ$C isotherm) is found to vary by 90 m from 435 to 525 m (Fig. 9). Of greater physical significance is the maximum depth where interstitial ice is present in permafrost. Due to the freezing-point depression caused by interfacial, curvature, pressure, and solute effects, the base of ice-bearing permafrost (B-IBPF) is located 20–27 m above the $P_d$ throughout the simulation. During the most recent ice-age cycle, the B-IBPF and $P_d$ both reached their greatest depths at $\sim 14$ ka, a delay of 10 kyr from the Last Glacial Maximum. Since then, these interfaces have been steadily rising. With the conditions of this simulation, the B-IBPF is currently located at 431 m in a silty claystone, about 22 m below the shallowest depth projected to have occurred following the last interglacial, while the base of permafrost $P_d$ is currently at 467 m in the underlying sandstone unit. Both interfaces are predicted to be rising about 1 cm yr$^{-1}$ at present, a rate that may be detectable with a carefully designed experiment.





**Figure 9.** Upper boundary condition ($T_s$) used to explore the response of a sedimentary sequence (Fig. 8) to surface-temperature changes over the last ice-age cycle. Also shown are the depths for the base of permafrost ($P_d$) and the base of ice-bearing permafrost (B-IBPF).

As the simulation confirms, the volume fraction of ice $\phi_i$ depends strongly on lithology (Fig. 10). This leads to zones with relatively high ice contents in coarse-grained sediments as is often observed in electrical resistivity geophysical logs (Hnatiuk and Randall, 1977; Osterkamp and Payne, 1981). An interesting facet of this behavior is that a pocket of ice-rich sandstone can occur below a fine-grained unit that has little or no ice (e.g., the silty claystone above the sandstone at 450 m in Fig. 10). Except

5   near the B-IBPF, the ice content in coarse-grained materials appears to be relatively stable. In contrast, the volume fractions of ice $\phi_i$ and unfrozen water $\phi_\ell$ are much more dynamic in fine-grained materials over ice-age cycles due to their greater temperature sensitivity. In the upper two-thirds of the permafrost zone, up to 15% of the water within silty claystones and limestones converts between ice and unfrozen water over ice-age cycles; the percentages are much higher in the lower third of the permafrost zone. Due to the phase change of water, a high heat capacity zone occurs just above the B-IBPF (Fig. 11). This

10  zone, which tracks the B-IBPF over time, is roughly 100 m thick. Pockets of elevated heat capacity also occur in fine-grained materials closer to the surface, especially during periods affected by the warm interglacials. Thermal conductivity variations over an ice-age cycle are also primarily driven by the melting and refreezing of pore ice. Hence, conductivity variations are greatest near the B-IBPF where conductivity changes of ±15% occur (Fig. 12). In the upper two-thirds of the permafrost zone, thermal conductivity variations are greater in the fine-grained materials, at least on a percentage basis.





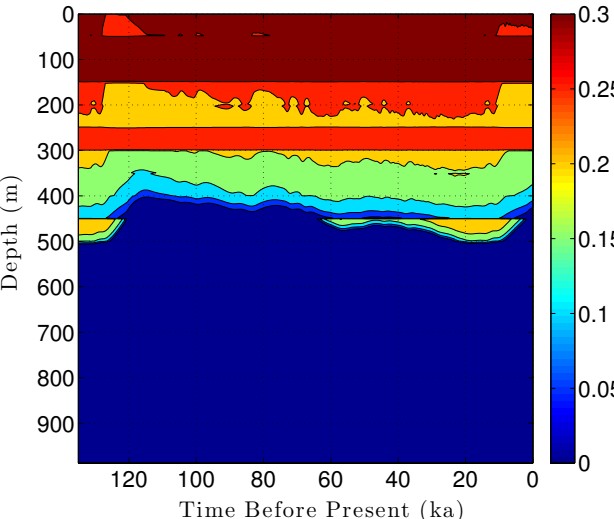

**Figure 10.** Ice content $\phi_i$ variations over the last ice-age cycle within the example sedimentary sequence (Fig. 8). Unfrozen water $\phi_\ell$ variations mirror the ice content fluctuations.

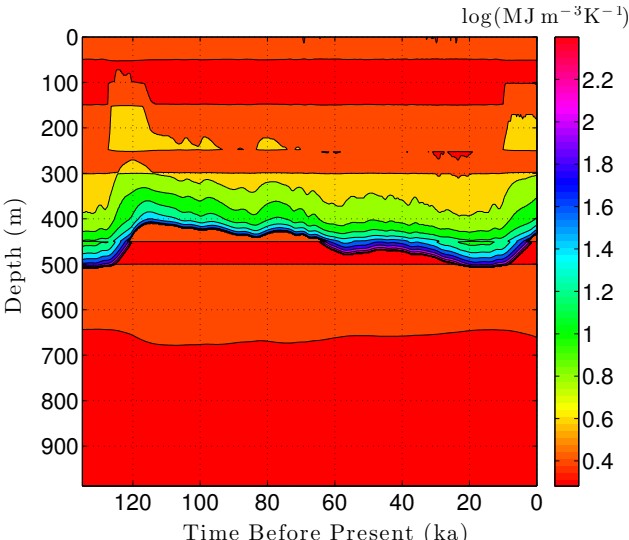

**Figure 11.** Volumetric heat capacity $C$ variations over the last ice-age cycle within the example sedimentary sequence (Fig. 8). The color mapping corresponds to $\log(C)$.





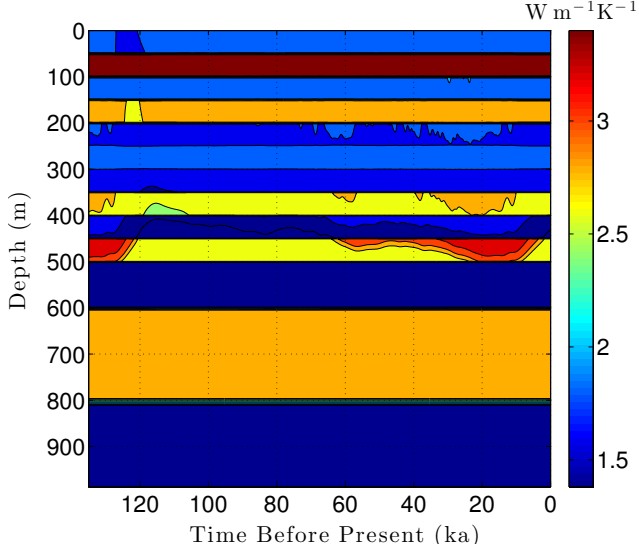

**Figure 12.** Bulk thermal conductivity $k$ variations over the last ice-age cycle within the example sedimentary sequence (Fig. 8).

## 4.2 Permafrost response to drilling a deep borehole

We now return to the state of the system in 1980 as determined in the previous example. By 1980, temperatures in the upper
100 m of the sedimentary sequence have warmed in response to the 2.5 K surface warming since the termination of the Little Ice
Age ($\sim$ 1850). At greater depths, temperatures still reflect conditions earlier during the Holocene and the Wisconsin Glacial

Period. With this initial state, we consider the drilling of a 3 km deep borehole through the example sedimentary sequence
over a 60 day period. Drilling fluids pumped into the 30 cm diameter hole at 30°C interact thermally with the drill pipe and
surrounding rock as they circulate to the bottom of the hole and back to the surface. As a result of the drilling processes, rocks
surrounding the hole warm throughout the permafrost zone. The degree of warming depends on both depth and time as the
drill bit advances into the warmer rocks below (Clow, 2015). Figure 13 shows the evolution of temperature changes $\Delta T_a$ at the

borehole wall during drilling based on the Szarka and Bobok (2012) wellbore model. This thermal drilling disturbance is used
in conjunction with the initial temperature field to establish the boundary condition at the borehole wall over the 60 day drilling
period. To complete the setup for a 2-D cylindrical simulation, the problem domain is extended far enough from the borehole
(40 m) that the radial heat flux at that boundary can be set to zero. The heat flux on the lower boundary is again 60 mW m$^{-2}$.

Running CVPM with the described initial and boundary conditions, the drilling disturbance is found to be great enough to

melt all of the permafrost ice within 1–2 m of the well by the time the borehole is completed on day 60 (Fig. 14). In this case,
borehole electrical resistivity measurements used to detect ice in permafrost must be able to penetrate at least 1.5 m of rock to
successfully detect ice. The exact location of the melting front is controlled partially by lithology, with it advancing further from
the borehole in fine-grained high-conductivity sediments (e.g., limestones) than in coarser grained low-conductivity mudrocks
such as siltstones. Sediment texture is a factor because it affects the volumetric ice content of a material in its undisturbed state.

Although the thermal disturbance due to drilling is greatest inboard of the melting front, a substantial disturbance also occurs





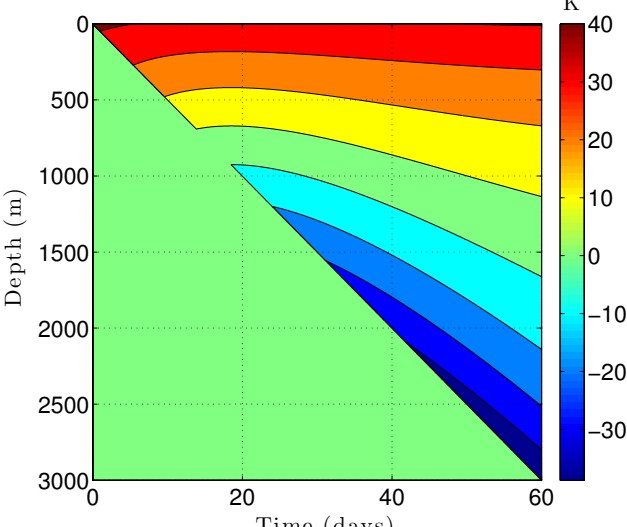

**Figure 13.** Temperature change $\Delta T_a$ at the borehole wall while drilling a 3 km borehole through the example sedimentary sequence (Fig. 8) over a 60 day period. This thermal disturbance is used to establish the boundary condition at the borehole wall during drilling.

beyond the front, particularly in the upper couple hundred meters of permafrost where undisturbed temperatures are $-7$ to $-9°$C (Fig. 15). Outboard of the melting front, the drilling disturbance extends further into the higher conductivity limestone and sandstone units than in the low conductivity mudrocks. Because of the effect of temperature on the thermal conductivity of minerals, ice, and water, and because of the conversion of ice to liquid water, the bulk thermal conductivity drops about

5   30% in the sedimentary units near the wellbore during drilling (Fig. 16). Attempts to infer the thermophysical properties of the sedimentary units from borehole temperature measurements using geophysical inverse methods must carefully consider these changes.



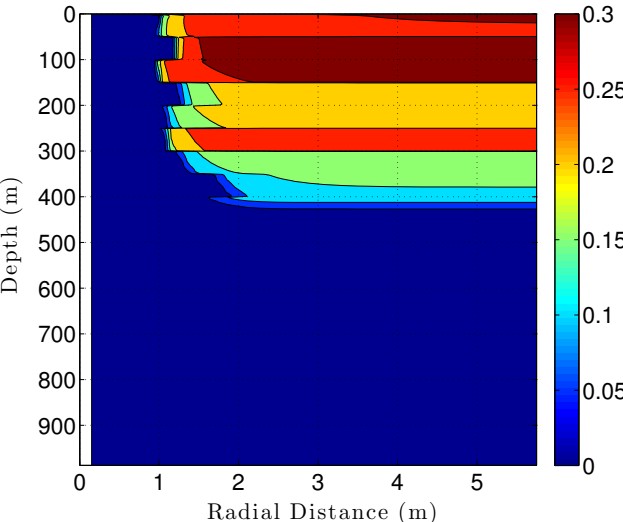

**Figure 14.** Volumetric ice content $\phi_i$ in the sedimentary sequence (Fig. 8) penetrated by a newly completed 3 km deep borehole. Radial distance is measured from the central axis of the borehole.

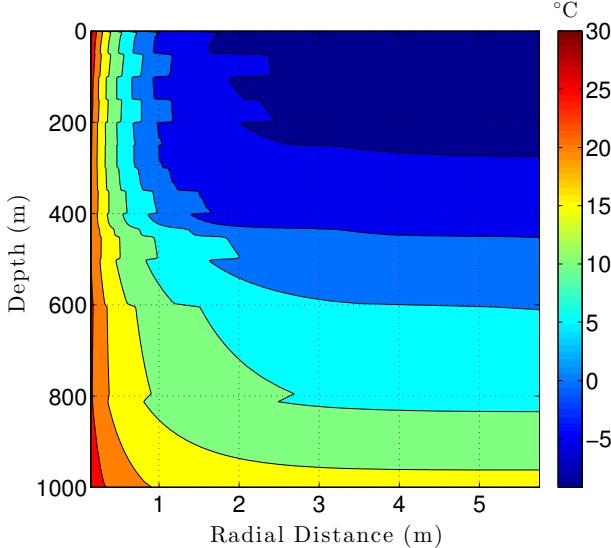

**Figure 15.** Temperatures in the sedimentary sequence (Fig. 8) penetrated by a newly drilled 3 km deep borehole (day 60).



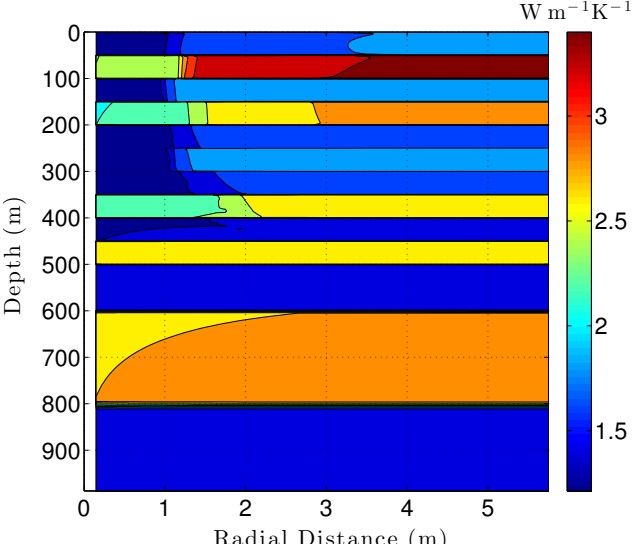

**Figure 16.** Bulk thermal conductivity $k$ in the sedimentary sequence (Fig. 8) penetrated by a newly drilled 3 km deep borehole (day 60).

### 4.3 Permafrost response to the formation of a lake

Shallow lakes are ubiquitous on the arctic coastal plains. In thermokarst areas, these lakes are constantly in transition, shrinking, enlarging, draining, and filling new depressions in response to changing temperatures and stream flows. The seasonal ice that forms on these lakes is categorized as 'bedfast' ice if it freezes solid to the bottom of the lake, 'floating' ice if some liquid remains beneath the ice throughout the winter, and 'intermittent' if it is bedfast some years and floating during others. Whether a lake is a bedfast-ice lake or a floating-ice lake depends on whether the maximum seasonal ice-cover thickness $Z_{ice}^{max}$ exceeds the depth of the lake. During the 1970s and 1980s, $Z_{ice}^{max}$ for lakes along the Beaufort Sea Coast of Alaska was $2.0 \pm 0.2$ m (Weeks et al., 1981; Arp et al., 2012). By 2010, the maximum seasonal ice thickness had decreased to about 1.5 m due to the warming climate in the region over the last few decades (Bieniek et al., 2014; Wang et al., 2017). Thus, 1.5 m deep lakes that would have been bedfast-ice lakes during the 1970s and 1980s would have transitioned to intermittent-ice lakes by 2010. Arp et al. (2012) recently provided lake-bed temperature data for seasonally ice-covered lakes near the Beaufort Sea Coast. Based on these data, lakes whose depth is 0.5–1.0 m less than $Z_{ice}^{max}$ have a mean-annual temperature $\sim 4$ K warmer than the surrounding tundra while intermittent-ice lakes are about 9 K warmer.

Here, we briefly explore the permafrost response over a 35 yr period to a 200 m wide lake instantaneously created on the Arctic Coastal Plain of Alaska in 1980. Initial conditions are the same as in the previous example. The lake is 1.5 m deep with a 100 m wide shallow (1.0 m deep) shelf lying in a small depression on the sedimentary sequence (Fig. 8). Similar to recent trends along the Beaufort Coast of Alaska (Wang et al., 2017; Clow, 2017), the tundra surrounding the lake is assumed to warm at 0.75 K decade$^{-1}$ over the simulation period from an initial surface temperature of $-8.5°$C. While the seasonal ice-cover is bedfast across the entire lake when it is first created, the ice-cover transitions to intermittent over the deeper section towards the



end of the simulation. Thus lake-bed temperatures in the deeper section, initially 4 K warmer than the surrounding tundra, warm until they are 9 K warmer than the tundra by year 35. The shelf remains 4 K warmer than the tundra throughout the simulation. Using the tundra and lake-bed temperatures as the upper boundary condition in CVPM (2-D cartesian mode), temperatures beneath the deeper portion of the lake become warm enough to melt all of the pore ice at the bed interface 19 years after the

5  lake is created (Fig. 17). Thereafter, the melting front propagates downward in the upper silty claystone unit at $\sim 19\,\mathrm{cm\,yr^{-1}}$ creating a thaw bulb in its wake (Fig. 18). Lateral migration of the melting front is much more modest, $\sim 4\,\mathrm{cm\,yr^{-1}}$ where the deep section adjoins the tundra and $\sim 6\,\mathrm{cm\,yr^{-1}}$ where it meets the shallow shelf. By the end of the simulation, a thaw bulb has not yet begun to form beneath the shelf. There are two ramifications of the growing thaw bulb beneath the deeper section of the lake: (1) Fine-grained materials such as silty clay lose much of their mechanical strength once they thaw. In this state, the

10 sides of the lake are much more vulnerable to erosion which may lead to eventual drainage of the lake. (2) Old carbon stocks stored in the previously frozen permafrost are likely to decompose in the anaerobic thaw bulb and contribute greenhouse gases to the atmosphere (Anthony et al., 2016).

## 5  Summary and conclusions

This paper presents the governing equations and numerical methods underlying the Control Volume Permafrost Model v1.1 which was designed to relax several of the limitations imposed by previous models. CVPM implements the nonlinear heat-transfer equations in 1-D, 2-D, and 3-D cartesian coordinates, as well as in 1-D radial and 2-D cylindrical coordinates. To

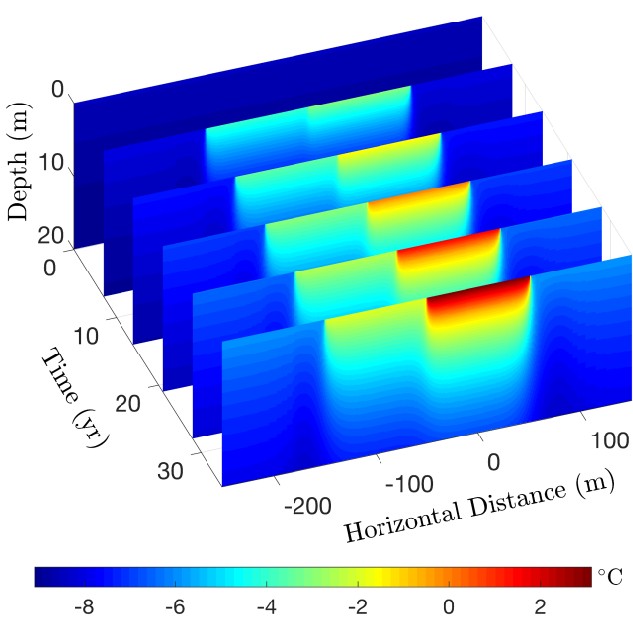

**Figure 17.** Simulated permafrost temperatures over a 35 yr period following the creation of a 200 m wide lake on a silty claystone unit. Depth is measured relative to the bottom of the deeper portion of the lake.



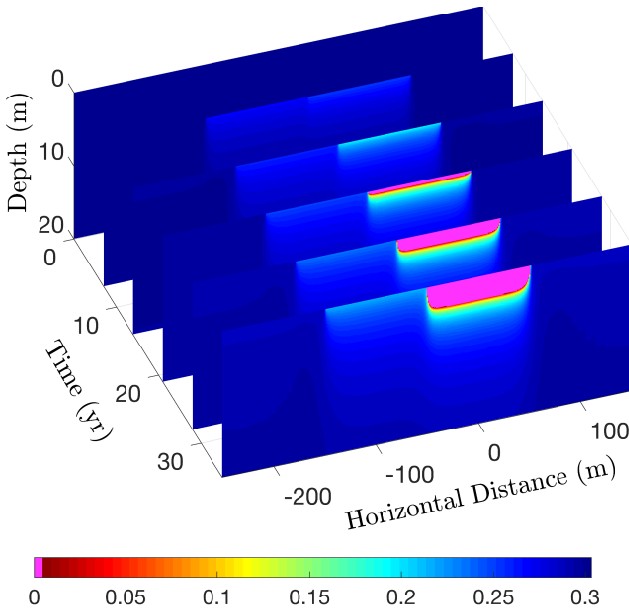

**Figure 18.** Volume fraction of ice $\phi_i$ over a 35 yr period following the creation of a 200 m wide lake. Magenta ($\phi_i = 0$) delineates the thaw bulb developing beneath the lake.

accommodate a diversity of geologic settings, a variety of materials can be specified within the modeling domain, including: organic-rich materials, sedimentary rocks and soils, igneous and metamorphic rocks, ice bodies, borehole fluids, and other engineering materials. Porous materials are treated as a matrix of mineral and organic particles with pores spaces filled with liquid water, ice, and air. Functions describing the temperature dependence of the specific heat $c_p$ and thermal conductivity $k$ are built into CVPM for a wide variety of rocks and minerals, water, air, and other substances. For porous materials, the

bulk thermal conductivity is found using the Brailsford and Major random 2-phase (matrix particles, pores) model while the conductivity of the pores themselves is found using a 3-phase (liquid water, ice, air) mixing model. This scheme allows the bulk thermal conductivity to be found for a wide range of porosities, water saturations ranging 0–100%, and different planetary atmospheres. In addition to the lattice-vibration term ($\rho c_p$), the volumetric heat capacity $C$ depends on a latent-heat term proportional to the change in liquid water content with temperature ($\partial \phi_\ell / \partial T$). At temperatures below 0°C, the unfrozen water

content is found using relationships from condensed matter physics that utilize physical quantities rather than non-physical empirical coefficients requiring calibration. Solute and pore pressure effects are included in the unfrozen water equations. As the amount of unfrozen water diminishes with cooling temperatures, pore-water solutes become increasingly concentrated in the remaining liquid water due to the low-solubility of most impurities in ice. To allow for the non-ideal behavior that occurs at high solute concentrations, the water activity $a_w$ is used to find the effect of solutes on the bulk freezing temperature of

water $T_f$. With this approach, solute concentrations up to the eutectic point are allowed. Pore pressure effects on $T_f$ are found in CVPM using either hydrostatic or lithostatic equations, whichever is more appropriate geologically. A radiogenic heat-





production term is also included to allow simulations to extend into deep permafrost and underlying bedrock. For the current version of CVPM, liquid water velocities are assumed to be small enough that the associated advective heat flux is negligible compared to the diffusive heat flux.

Numerical implementation of the governing equations is accomplished using the control-volume approach, allowing enthalpy fluxes to be exactly balanced at control-volume interfaces (e.g., at the interfaces between ice lenses, sedimentary units, bedrock, or a borehole casing). This approach was chosen because the expressions tend to be more accurate than with other methods near boundaries and where strong thermal or physical-property contrasts occur. Very large thermal-property contrasts generally occur near the water freezing point in permafrost. Despite the magnitude of the contrasts and the fact that the freez-

ing front typically migrates over time, the numerical scheme used in CVPM remains stable as long as the stability criterium (Eq. 40) is satisfied. CVPM can be run in either explicit, implicit, or fully implicit modes.

    CVPM has been designed for a wide range of scientific and engineering applications, and as an educational tool. The model is suitable for use at spatial scales ranging from centimeters to hundreds of kilometers and at timescales ranging from seconds to thousands of years. CVPM can be used over a broad range of depth, temperature, porosity, water saturation, and solute conditions on either the Earth or Mars. Through its modular design, CVPM can act as a stand-alone model, the physics package of a geophysical inverse scheme, or serve as a component within a larger earth modeling system that may include vegetation, surface water, snowpack, atmospheric or other models of varying complexity.

    One of the goals of CVPM was to eliminate the empirical equations typically used to predict the unfrozen water content at

5 temperatures below $0°C$, replacing them with condensed matter relationships containing quantities that both have a physical meaning and can be measured using relatively simple techniques. The physical quantities in the resulting equations tend to occur within reasonably narrow limits for the material categories defined in CVPM. Thus if measurements of the physical quantities are unavailable, it may be possible to estimate the behavior of permafrost in a region from a geologic description of the area.

*Code availability.* CVPM source code, test cases, examples, and user's guide are publicly available at the Community Surface Dynamics Modeling System repository at https://csdms.colorado.edu/wiki/Model:CVPM. CVPM v1.1 is implemented in the MATLAB programming language and is distributed under the GNU General Public License v3.0.

*Competing interests.* The author declares that he has no conflict of interest.

*Acknowledgements.* This work was supported by the U.S. Geological Survey through a grant from the Climate and Land Use Change
Program. Any use of trade, firm, or product names is for descriptive purposes only and does not imply endorsement by the U.S. Government. The author thanks the referees for their careful reviews and constructive suggestions which helped improve the manuscript.



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
