# Peer review of "CVPM 1.1: a flexible heat-transfer modeling system for permafrost"

_Geoscientific Model Development, 2018_

## Referee Comment (RC1) · Anonymous Referee #1 · 16 Jul 2018

**CVPM 1.1: a flexible heat-transfer modeling system for permafrost**
**By G.D. Clow**

This is the fundamental work describing in-depth modeling of the subsurface heat-flow in presence of permafrost that deserves to be published. The advantage of the CVPM model is that it is scalable from 1D to 3D and can be used in multiple coordinates including cartesian, radial and cylindrical. It accommodates diversity of the geologic settings for model setup and employs more physical approach for setting up the freezing depression point. Many existing permafrost models use empirically derived parameters to represent unfrozen water function, which is an important component responsible for correct temperature modeling and precision of the freeze/thaw moving boundary. However, from the current description is it not clear why and how the unfrozen water content used in CVPM is different from empirical derived unfrozen water coefficients, for example (a, b, and freezing point depression $T^*$) used in GIPL model (Nicolsky et al, 2009; Jafarov et al., 2012). It is hard to clearly differentiate these two approaches without direct benchmark comparison of these models. In addition, CVPM uses empirically derived coefficients in equations (6) and (7), somewhat similar to GIPL. It would be nice to add an example of the modeled versus observed ground temperatures from one or multiple permafrost monitoring stations from North Slope, Alaska (Wang et al., 2018). Do the coefficients from Table 1 fulfil the whole range of subsurface complexity to represent observed ground temperatures?

Below I have minor suggestions aimed to improve the clarity of the manuscript.

Figure 1. I suggest to include two more plots: b) $c_{pl}$ vs $T(k)$, and c) $c_{pi}$ vs $T(k)$ as a subplots for Figure 1, where current plot is a). Add a line illustrating liquid-liquid critical temperatures. Line 7, most material fall within the range 630-870 J/kgK. I suggest to add a bar to the Figure 1A that represent that range.

What is the main advantage using equations (6) and (7) in oppose to apparent heat capacity formulation? Coefficients in Table 1, where are they come from?

P6. L5. L10. It would be nice to have visual representation of the porous media and unfrozen water redistribution for spherical and cubic packing.

Figure 2. It is not clear according to which formula $phi_i$ is calculated. Equation (9) does not have d or $T_f$ in it.

P7. L5. Pure water freezes at 273.15K. I am not sure where 273.16 and triple point pressure come from. Need a citation.

P9. Figure 3B. How the derivative was calculated (numerically or analytically)? I suggest to add an equation of the derivative.

P10. Eqn. 17, for consistency with eqn. 19, should it be $k_{BM2}$?

Table 5. It would be nice to list below the table the meaning of each notation mean.

An example shown on Figure 9. What was the time step?

Figure 10. Colorbar does not have notation.

Figure 13. What are the initial and boundary condition?

P24. L6-7. Need a citation.
Section 4.3. What are the initial (IC) and boundary condition (BC)? What type of ground material and layering was used?

P28. L10 "bulk thermal conductivity is found…". Not clear, what does it mean "found"? Inversely estimated?

I felt that for all examples the domain setup including IC and BC was skipped. Since this the modeling paper, I suggest to add a bit more details on model setup, grid (mesh), thermal, hydrological and other parameters setup. What type of data do the model need for initialization? What is the model driver data? What is model subsurface setup (soil texture, layering, thermal conditions and so on).

**References**

1. D. J. Nicolsky, V. E. Romanovsky and G. G. Panteleev, Estimation of soil thermal properties using in-situ temperature measurements in the active layer and permafrost, Cold Reg. Sci. Technol., 55 (2009), 120–129.
2. Jafarov, E. E., Marchenko, S. S., and Romanovsky, V. E.: Numerical modeling of permafrost dynamics in Alaska using a high spatial resolution dataset, The Cryosphere, 6, 613-624, https://doi.org/10.5194/tc-6-613-2012, 2012.
3. Wang, K., Jafarov, E., Schaefer, K., Overeem, I., Romanovsky, V., Clow, G., Urban, F., Cable, W., Piper, M., Schwalm, C., Zhang, T., Kholodov, A., Sousanes, P., Loso, M., and Hill, K.: A synthesis dataset of permafrost-affected soil thermal conditions for Alaska, USA, Earth Syst. Sci. Data Discuss., https://doi.org/10.5194/essd-2018-54, in review, 2018.

---

## Referee Comment (RC2) · Anonymous Referee #2 · 27 Aug 2018

This nice paper describes a new and comprehensive heat-transfer modeling system for permafrost. The model implements a set of heat-transfer physics that are more detailed than in most commonly used large-scale permafrost models. It appears to be highly flexible and therefore applicable to a range of permafrost research problems across a range of spatial scales and timescales. This capacity is nicely demonstrated with three applications that span an impressive array of timescales and research topics (permafrost thermal evolution from 255ka years ago to present to examine permafrost evolution over ice age cycles; a 60 day detailed simulation of the impact of a bore hole drilling operation; and permafrost response to formation of a lake). The model is designed to work for a range of geologic settings as well.

The paper is clear and well-written and the model is described in sufficient detail to

really understand how and why the model was constructed as it was. Overall, I find very little to criticize and I find the paper suitable for publication, essentially in its current form. The model should be an excellent resource for the permafrost research community.

A couple minor points.

1. Maybe I missed it, but I think it would be helpful if the author could explain in a bit more detail what is needed to force the model. Is it just surface temperature?

2. Are soil and rock water amounts prescribed and not allowed to change? There isn't any description of soil hydrology so that would suggest that that is the case. If so, then if one wanted to couple this into a large-scale permafrost or Earth System model, it would just replace the heat-transfer solution, and the host model would calculate water flow through the soil and sediment? Would there be any impediments to doing this?

3. Could the CVPM be coupled with a surface energy balance model?

4. Along similar lines, the author notes that the CVPM does not represent vegetation, snow, surface water, etc. This makes me wonder how the example simulations were executed. Is the model forced with ground surface temperature, i.e., the temperature from beneath the snow.

5. Would maybe be helpful to indicate what the timestep is for each of the example applications with a brief description of the implications of the timestep. If, for example, the timestep is annual or longer for the 255kyr simulation, then this obviously implies that these simulations cannot be used to track active layer thickness. If the timestep is shorter than annual, then how does one derive the forcing timeseries, which obviously isn't resolved. (Apologies if these are stupid questions, I don't usually think on timescales that long!).
* * *

---

## Author Comment (AC1) · 15 Oct 2018

**Author Response to Reviewer 1**

-AC: I thank the reviewer for taking the time to review this paper and for the constructive comments which have helped improve the clarity of the manuscript. Author responses (AC) to the reviewer's comments are provided in blue text below each of the comments.

**General comments:**

This is the fundamental work describing in-depth modeling of the subsurface heat-flow in presence of permafrost that deserves to be published. The advantage of the CVPM model is that it is scalable from 1D to 3D and can be used in multiple coordinates including cartesian, radial and cylindrical. It accommodates diversity of the geologic settings for model setup and employs more physical approach for setting up the freezing depression point.

Many existing permafrost models use empirically derived parameters to represent unfrozen water function, which is an important component responsible for correct temperature modeling and precision of the freeze/thaw moving boundary. However, from the current description is it not clear why and how the unfrozen water content used in CVPM is different from empirical derived unfrozen water coefficients, for example (a, b, and freezing point depression T*) used in GIPL model (Nicolsky et al, 2009; Jafarov et al., 2012). It is hard to clearly differentiate these two approaches without direct benchmark comparison of these models. In addition, CVPM uses empirically derived coefficients in equations (6) and (7), somewhat similar to GIPL.

-AC: In my understanding, the disadvantages of using an empirical function to represent the unfrozen-water content such as is done by GIPL and many other permafrost models are two-fold: (a) The empirical function coefficients (e.g., $a$ and $b$) must be determined from detailed temperature and unfrozen-water measurements on geologic samples or inferred from detailed in-situ measurements using geophysical inverse methods. However, for most of the earth's permafrost we either lack geologic samples on which to make the measurements or in-situ measurements of sufficient detail that the coefficients can be inferred from an inversion. The enormous range of $a, b$ values reported in the literature suggests we cannot know their values with any certainty in situations where we lack samples or the necessary in-situ measurements (e.g., for permafrost deeper than $\sim 50\,\mathrm{m}$ on the earth or anywhere on Mars). An additional complication is that the unfrozen-water content $\phi_\ell$ is very sensitive to impurities. Thus, without detailed solute measurements to allow the separation of the solute effects from the ground's textural effects, the $a, b$ values determined from a geologic sample or an inversion will only be valid for a particular lithologic/solute-concentration combination. (b) An empirical unfrozen-water function is only valid over the range of temperatures for which it has been experimentally determined. There is no guarantee that the form of the function or the empirical coefficients are valid beyond this limited range. The work of Cahn et al. (1992) and others suggests that the empirical unfrozen-water functions used by most permafrost models today are too simple, particularly near the bulk freezing temperature $T_f$. The need to model temperatures and unfrozen-water content in permafrost situations for which we are unable to establish an empirical $\phi_\ell$ function with any certainty (e.g., permafrost deeper than $\sim 50\,\mathrm{m}$ on the earth, anywhere on Mars, or at temperatures beyond which an empirical function might be valid) provided the motivation for the physically-based approach adopted by the CVPM model. In this approach, the unfrozen-water content is determined using a physics-based equation that depends of the particle (or pore) sizes $r$ and the temperature below the bulk freezing temperature $T_f$. The advantage of this approach is that the particle size can be determined from deep geophysical logs, or barring that, estimated from a description of the geologic units since those units are often defined by particle size (e.g., a fine silt consists of particles in the $8$–$16\,\mu\mathrm{m}$ size range). The temperature below the freezing point is determined by the model. There is an empirical coefficient in CVPM's unfrozen-water relationship (Eq. 10) that is in some way equivalent to the $a, b$ coefficients found in most empirical unfrozen-water functions. This coefficient is the

interfacial melting parameter $\lambda$. However, $\lambda$ appears to be much more tightly constrained for most geologic materials than are the empirical coefficients $a, b$; preliminary determinations find that $\lambda$ varies by perhaps 10–20% across a wide range of materials. Equations (6) and (7) define the specific heat of liquid water ($c_{p\ell}$) and of ice ($c_{pi}$) relationships and thus are not directly related to the unfrozen water content.

It would be nice to add an example of the modeled versus observed ground temperatures from one or multiple permafrost monitoring stations from North Slope, Alaska (Wang et al., 2018).

-AC: I agree that a comparison of modeled versus observed ground temperatures is important. However, given the uncertainty of the thermophysical properties at the monitoring stations, such a comparison would require a great deal of additional discussion. As such, this will left as a topic for a future paper.

Do the coefficients from Table 1 fulfill the whole range of subsurface complexity to represent observed ground temperatures?

-AC: The coefficients in Table 1 pertain to the equations for the specific heat of liquid (unfrozen) water and of ice (Eqs 6–7). These forms of water are the same regardless of the geologic materials in which they are found. Thus, they should remain valid over the whole range of subsurface complexity.

**Minor comments:**

[1] Figure 1. I suggest to include two more plots: b) cpl vs T(k), and c) cpi vs T(k) as a subplots for Figure 1, where current plot is a). Add a line illustrating liquid-liquid critical temperatures. Line 7, most material fall within the range 630-870 J/kgK. I suggest to add a bar to the Figure 1A that represent that range.

-AC: A second plot (Fig 1b) has been added showing $c_{p\ell}(T)$ and $c_{pi}(T)$ in more detail. A line illustrating the location of the liquid–liquid critical point has been added to Fig 1B and a vertical bar showing the range of specific heats for most minerals has been added to Fig 1A.

[2] What is the main advantage using equations (6) and (7) in oppose to apparent heat capacity formulation? Coefficients in Table 1, where are they come from?

-AC: Calculation of the lattice-vibration component ($\rho c_p$) of the volumetric heat capacity requires values for the specific heat of liquid water ($c_{p\ell}$) and of ice ($c_{pi}$) over a broad range of temperatures. Equations (6) and (7) provide the necessary $c_{p\ell}(T)$ and $c_{pi}(T)$ relationships. The $c_{p\ell}$ coefficients listed in Table 1 were determined using a least-squares fit to the data reported by Angell et al. (1982) and the IAPWS (2008) as described on Pg 4, lines 14–16. The $c_{pi}$ coefficients were derived by putting Yen's original empirical relationship in a slightly different form.

[3] P6. L5. L10. It would be nice to have visual representation of the porous media and unfrozen water redistribution for spherical and cubic packing.

-AC: For a visual representation of the unfrozen water distribution, I refer interested readers to the following paper: Cahn, J.W., Dash J.G., and Fu, H.: Theory of ice premelting in monosized powders, J. Crystal Growth, 123, 101–108, 1992.

[4] Figure 2. It is not clear according to which formula phii is calculated. Equation (9) does not have d or Tf in it.

-AC: The volume fraction of liquid water $\phi_\ell$ in Fig. 2 is calculated using Eq. (9) with the radius $r$ being half the particle diameter $d$ and $\Delta T = T_f - T$. Additional text has been added to the caption of Fig. 2 to clarify how $\phi_\ell$ was calculated.

[5] P7. L5. Pure water freezes at 273.15K. I am not sure where 273.16 and triple point pressure come from. Need a citation.

-AC: Although not explicitly stated, the CVPM model utilizes the ITS-90 temperature scale. The triple point of pure water is the most important defining point on this scale. From Kittel (1969), Guildner et al. (1976), and Nicholas and White (2001), the triple point of water occurs at temperature $T_f^\star =$273.16 K and pressure $P_{tp} = 611.657$ Pa. From Eq. (16), the freezing point depression at standard sea level pressure (101.3 kPa) is $\theta_p = 0.01$ K. Thus at sea level, CVPM correctly predicts pure air-saturated water will freeze at $T_f = 273.15$ K (Eq. 12). In portions of the Tibetan Plateau exceeding elevations of 5000 m, surface pressures are only $\sim 55$ kPa. The corresponding freezing point depression is $\theta_p \simeq 0.005$ K putting the expected freezing temperature for air-saturated water at $T_f = 273.155$ K. The above citations have been added to the manuscript.

[6] P9. Figure 3B. How the derivative was calculated (numerically or analytically)? I suggest to add an equation of the derivative.

-AC: The derivative $\partial\phi_\ell/\partial T$ is calculated numerically. For computational efficiency, a table of $\phi_\ell(T)$ values is constructed at the beginning of a simulation run. All subsequent requests for $\phi_\ell(T)$ are found by interpolating values in the table. The derivative at temperature $T$ is found using a simple difference operation using two calls to the interpolation table over a temperature span of $\pm 1\,\mu$K.

[7] P10. Eqn. 17, for consistency with eqn. 19, should it be kBM2?

-AC: I agree the notation is a bit confusing. To help alleviate this, the notation $k_{BM3}$ in Eq. (19) has been replaced by $k_{cx}$. The new notation connects better with the continuous phase terms ($k_{c\ell}$, $k_{ci}$, $k_{ca}$) appearing on lines 14–15 (Pg 11) and in Eq. (20). The simple notation for the bulk thermal conductivity ($k$) in Eq. (17) is retained because of this term's fundamental importance.

[8] Table 5. It would be nice to list below the table the meaning of each notation mean.

-AC: A description of the parameter notation has been added to the caption of Table 5.

[9] An example shown on Figure 9. What was the time step?

-AC: The computational time step in this example is 20 years. This was deemed sufficient given our interest in demonstrating the evolving thermal properties well below the surface in this example and in permafrost thickness changes. The time step is now stated in the text along with the vertical grid spacing.

[10] Figure 10. Colorbar does not have notation.

-AC: Figure 10 shows ice content $\phi_i$ variations. Because $\phi_i$ is dimensionless, no notation is included with the colorbar to retain consistency with the other figures.

[11] Figure 13. What are the initial and boundary condition?

-AC: The initial condition is given by the state of the sedimentary sequence in 1980 as determined by the example in Section 4.1 (all the state variables from the Section 4.1 experiment were output to a file at year 1980 to be used as the initial condition for the Section 4.2 and 4.3 experiments). The boundary condition at the borehole wall during the 60-day simulation in this example is given by adding the thermal drilling disturbance shown in Fig. 13 to the initial temperature field. The initial condition and boundary condition are described on Pg 23, lines 2–13. I have reworded this section in a few places to help clarify the initial and

boundary conditions. Text has also been added describing the 2-D cylindrical grid and the computational time step.

[12] P24. L6-7. Need a citation.

-AC: A citation has been added to the manuscript.

[13] Section 4.3. What are the initial (IC) and boundary condition (BC)? What type of ground material and layering was used?

-AC: The initial condition is the same as for the example in Section 4.2 (see AC response, comment 11). The upper boundary condition (described on Pg 26, lines 17–19 and Pg 27, lines 1–2) varies according to location. Beneath the tundra, temperatures were assumed to warm from the initial 1980 surface temperature ($-8.5$°C) at $0.75$ K decade$^{-1}$. Beneath the deeper portion of the lake, lake-bed temperatures were initially 4 K warmer than the surrounding tundra. They then warmed until they were 9 K warmer than the tundra by 2015. Temperatures beneath the shallow shelf were 4 K warmer than the tundra for the entire simulation. Temperatures at the 1.5 m depth beneath the tundra, shelf, and deeper portion of the lake provided the upper BC for the simulation. A heat-flux BC was prescribed on the lower boundary ($q_b = 60$ mW m$^{-2}$), taken to occur at the 300 m depth in this example. The problem domain was extended laterally far enough beyond the lake that the heat flux across the outer boundaries could be set to zero. The ground material and layering was the same as the previous two examples (Pg 19, lines 13–16). Portions of Section 4.3 have been reworded to clarify the initial conditions, boundary conditions, and material properties. Text has been added describing the 2-D cartesian grid and the computational time step.

[14] P28. L10 'bulk thermal conductivity is found..'. Not clear, what does it mean 'found'? Inversely estimated?

-AC: The bulk thermal conductivity is 'found' using the Brailsford and Major random 2-phase relationship described in Section 2.3 (Eq. 17). This 2-phase relationship requires the conductivity of the matrix and the pores which are given by Eqs (18) and (19), respectively. The conductivity of the pores, in turn, depends on the thermal conductivity of liquid water, ice, and air which are described by Eqs (21) - (23). The thermal conductivity of the all physical components (matrix, liquid water, ice, air) are parameterized as described in Section 2.3. Essentially all that is needed to find the bulk thermal conductivity is then the temperature and the volume fractions. The wording in this section has been revised slightly to help clarify how the bulk thermal conductivity is found.

[15] I felt that for all examples the domain setup including IC and BC was skipped. Since this the modeling paper, I suggest to add a bit more details on model setup, grid (mesh), thermal, hydrological and other parameters setup. What type of data do the model need for initialization? What is the model driver data? What is model subsurface setup (soil texture, layering, thermal conditions and so on).

-AC: More details have been added to Sections 4.1–4.3 about the initial conditions and boundary conditions (see AC response for comments 11 and 13), the spatial grid, and the computation time step. The subsurface setup (lithology, thermophysical parameters) are now more explicitly stated. The *type* of data needed for model initialization are those found in Table 5 (described in more detail in the CVPM Version 1.1 Modeling System User's Guide available at: https://csdms.colorado.edu/wiki/Model:CVPM). Changing boundary conditions are all that's needed to drive the model. Additional text has been added to Section 3 clarifying the boundary conditions.

---

## Author Comment (AC2) · 15 Oct 2018

**Author Response to Reviewer 2**

-AC: I thank the reviewer for taking the time to review this paper and for the constructive comments which have helped improve the clarity of the manuscript. Author responses (AC) to the reviewer's comments are provided in blue text below each of the comments.

**General comments:**

This nice paper describes a new and comprehensive heat-transfer modeling system for permafrost. The model implements a set of heat-transfer physics that are more detailed than in most commonly used large-scale permafrost models. It appears to be highly flexible and therefore applicable to a range of permafrost research problems across a range of spatial scales and timescales. This capacity is nicely demonstrated with three applications that span an impressive array of timescales and research topics (permafrost thermal evolution from 255 ka years ago to present to examine permafrost evolution over ice age cycles; a 60 day detailed simulation of the impact of a borehole drilling operation; and permafrost response to formation of a lake). The model is designed to work for a range of geologic settings as well.

The paper is clear and well-written and the model is described in sufficient detail to really understand how and why the model was constructed as it was. Overall, I find very little to criticize and I find the paper suitable for publication, essentially in its current form. The model should be an excellent resource for the permafrost research community.

-AC: Thank you for the positive comments.

**Minor comments:**

[1] Maybe I missed it, but I think it would be helpful if the author could explain in a bit more detail what is needed to force the model. Is it just surface temperature?

-AC: CVPM is designed, as much as possible, to be a general-purpose numeral heat-transfer solver. All that is required to force the model are the boundary conditions on the edges of the problem domain. With the current version, either the temperature (Dirichlet condition) or the heat flux (Neumann condition) must be specified on each of the boundaries. These conditions are allowed to vary both spatially and temporally along the edges. For example with a 2-D problem, the temperature can be prescribed on the upper and lateral boundaries while the heat flux is prescribed on the lower boundary. Alternatively, the heat flux can be specified on one or both of the lateral boundaries or the temperature specified on the lower boundary. For the 2-D problem in Section 4.2, the model was forced by prescribed temperatures that varied with depth and time along the borehole wall (inner boundary). Temperatures were also prescribed on the upper boundary while heat fluxes were specified on the outer and lower boundaries. Although the upper, lower, and outer boundary conditions remained fixed, they too could have varied with position and time. For the 2-D problem in Section 4.3, the model was forced on the upper boundary with prescribed temperatures that varied with horizontal distance and time while prescribed heat flux conditions were specified on the other three boundaries. The text in Sections 3 and 4 has been revised to help clarify the boundary conditions. Heat exchange across a boundary by convection (Robin condition) will be included in a future version of CVPM.

[2] Are soil and rock water amounts prescribed and not allowed to change? There isn't any description of soil hydrology so that would suggest that that is the case. If so, then if one wanted to couple this into a large-scale permafrost or Earth System model, it would just replace the heat-transfer solution, and the host model would calculate water flow through the soil and sediment? Would there be any impediments to doing this?

-AC: The current version of CVPM does not allow the rock and soil water amounts to change with time,

although this is being considered for a future version. To couple the current version (v1.1) with a large-scale permafrost or Earth System Model, I would recommend coupling CVPM with a hydrologic model that calculates water flow across the surface and with an active-layer model that calculates heat and fluid flow through the active-layer sediments. As the control volume method used by CVPM was originally developed for problems in computational fluid dynamics, CVPM potentially can be modified to include the advective heat and fluid-flow terms. The required modifications would be fairly substantial but it would eliminate the need to couple CVPM to a separate active-layer model in cases where fluid flow within the active layer is deemed important.

[3] Could the CVPM be coupled with a surface energy balance model?

-AC: Yes, CVPM is designed in a modular form to facilitate coupling with surface energy balance and other models.

[4] Along similar lines, the author notes that the CVPM does not represent vegetation, snow, surface water, etc. This makes me wonder how the example simulations were executed. Is the model forced with ground surface temperature, i.e., the temperature from beneath the snow.

-AC: The model is generally forced at the highest level in the ground where advective heat transfer is small compared to the diffusive heat transfer. This level is safely taken at the top of permafrost. During winter in frozen terrain, the upper boundary can be taken a bit higher at the ground–snow interface. During the summer, it can be taken at the base of the active layer (top of permafrost), or at the ground–air interface if the heat flux associated with any fluid flow in the active layer is small. For the examples in Sections 4.1 and 4.2, the upper boundary was taken at the top of permafrost, about 30 cm below the surface. At the scale the model was being run at, the upper 30 cm was completely unresolved in these examples. For the example in Section 4.3, the upper boundary was taken 1.5 m below the surface, corresponding to the lake bed in its deepest section. This allowed the model to be driven by lake-bed temperatures. The model forcing used in the examples is now described in more detail in the manuscript.

[5] Would maybe be helpful to indicate what the timestep is for each of the example applications with a brief description of the implications of the timestep. If, for example, the timestep is annual or longer for the 255kyr simulation, then this obviously implies that these simulations cannot be used to track active layer thickness. If the timestep is shorter than annual, then how does one derive the forcing timeseries, which obviously isn't resolved.

-AC: For the ice-age cycle example in Section 4.1, the computational time step was 20 years. This was deemed sufficient since the focus of this simulation was on exploring the response of permafrost well below the surface, i.e., well below the active layer and any seasonal effects. For the drilling disturbance simulation (Section 4.2), the time step was set to 0.2 days to resolve the rapid temperature changes that occur near the advancing drill bit. For the lake response simulation (Section 4.3), the computational time step was set to 0.1 years. Although the upper boundary condition used to force the model in this case did not include any seasonal effects, a time step this short was required to satisfy the numerical stability condition (Eq. 40) with the fine spatial grid. If the user has a temperature timeseries with subannual variations, there is no impediment to running CVPM with a subannual computational time step. The time step is completely flexible as long as the numerical stability condition (Eq. 40) is satisfied. The time step and its implications for the examples is now described in more detail in the manuscript.